# Qualitative and Quantitative Analysis of Atmospheric Organosulfates in Centreville, Alabama

Anusha P. S. Hettiyadura[1], Thilina Jayarathne[1], Karsten Baumann[2], Allen H. Goldstein[3,4], Joost A. de Gouw[5], Abigail Koss[5], Frank N. Keutsch[6], Kate Skog[7] and Elizabeth A. Stone[1]

[1]Department of Chemistry, University of Iowa, Iowa City, IA, USA
[2]Atmospheric Research & Analysis, Inc., Cary, NC, USA
[3]Department of Environment Sciences, Policy and Management, University of California, Berkeley, CA, USA
[4]Department of Civil and Engineering, University of California, Berkeley, CA, USA
[5]Earth Systems Research Laboratory, Chemical Science Division, National Oceanographic and Atmospheric Administration,
Boulder, CO, USA
[6]School of Engineering and Applied Sciences, Department of Chemistry and Chemical Biology, Harvard University, Cambridge, MA, USA
[7]University of Wisconsin – Madison, Department of Chemistry, Madison, WI, USA

*Correspondence to*: Elizabeth A. Stone (betsy-stone@uiowa.edu)

**Abstract.** Organosulfates are components of secondary organic aerosols (SOA) that form from oxidation of volatile organic compounds (VOC) in the presence of sulfate. In this study, the composition and abundance of organosulfates were determined in fine particulate matter ($PM_{2.5}$) collected from Centreville, AL during the Southern Oxidant and Aerosol Study (SOAS) in summer 2013. Six organosulfates were quantified using hydrophilic interaction liquid chromatography (HILIC) with triple
quadrupole mass spectrometry (TQD) against authentic standards. Among these, the three most abundant species were glycolic acid sulfate ($0.5 – 52.5$ ng m$^{-3}$), lactic acid sulfate ($0.5 – 36.7$ ng m$^{-3}$) and hydroxyacetone sulfate ($0.5 – 14.3$ ng m$^{-3}$). These three species were strongly inter-correlated, suggesting similar precursors and/or formation pathways. Further correlations with sulfate, isoprene and isoprene oxidation products indicate important roles for these precursors in organosulfate formation in Centreville. Positive filter sampling artifacts associated with these organosulfates due to gas adsorption or reaction of gas
phase precursors of organosulfates with sulfuric acid were assessed for a subset of samples and were less than 7.8 % of their $PM_{2.5}$ concentrations. Together, the quantified organosulfates accounted for < 0.3 % of organic carbon mass in $PM_{2.5}$. To gain insights to other organosulfates in $PM_{2.5}$ collected from Centreville, semi-quantitative analysis was employed by way of monitoring of characteristic product ions of organosulfates ($HSO_4^-$ at *m/z* 97 and $SO_4^{-\bullet}$ at *m/z* 96) and evaluating relative signal strength by HILIC-TQD. Molecular formulas of organosulfates were determined by high-resolution time-of-flight (ToF) mass
spectrometry. The major organosulfate signal across all samples corresponded to 2-methyltetrol sulfates, which accounted for $42 – 62$ % of the total bisulfate ion signal. Whereas, glycolic acid sulfate, the most abundant organosulfate quantified in this study, was $0.13 - 0.57$ % of the total bisulfate ion signal. Precursors of *m/z* 96 mainly consisted of nitro-oxy organosulfates. Organosulfates identified were mainly associated with biogenic VOC precursors, particularly isoprene and to a lesser extent monoterpenes and 2-methyl-3-buten-2-ol (MBO). While a small number of molecules dominated the total organosulfate signal,

a large number of minor species were also present. This study provides insights to the major organosulfate species in the Southeastern US, as measured by tandem mass spectrometry that should be targets for future standard development and quantitative analysis.

# 1 Introduction

Atmospheric fine particulate matter ($PM_{2.5}$; particles $\leq 2.5$ µm in aerodynamic diameter) adversely affects human health (Valavanidis et al., 2008; Anderson et al., 2011; Kim et al., 2015) and influences the Earth's climate via direct and indirect radiative forcing (Novakov and Penner, 1993; Haywood and Boucher, 2000). A significant fraction of $PM_{2.5}$ organic matter is secondary in origin (Zhang et al., 2011), and forms by atmospheric oxidation reactions of volatile organic compounds (VOC) and partitioning of reaction products to the aerosol phase (Hallquist et al., 2009). Laboratory studies have shown increase of secondary organic aerosol (SOA) formation during oxidation of biogenic VOC in the presence of acidic sulfate (Surratt et al., 2007b; Surratt et al., 2010; Surratt et al., 2008; Surratt et al., 2007a; Liao et al., 2015). Among those SOA are organosulfates, which are mainly produced from acid-catalyzed particle-phase reactions of gaseous oxidation products such as epoxides (Lin et al., 2012) and hydroperoxides (Mutzel et al., 2015). Important precursors to organosulfates have been identified through a combination of field and laboratory studies, and include biogenic VOC such as isoprene (Surratt et al., 2007b), monoterpenes (Iinuma et al., 2009), sesquiterpenes (Chan et al., 2011), 2-methyl-3-buten-2-ol (MBO) (Zhang et al., 2012a) and 3-Z-hexenal (Shalamzari et al., 2014). Thus, organosulfates may be useful markers for sulfate-influenced biogenic SOA.

$PM_{2.5}$ mass in the Southeastern (SE) US is dominated by sulfate and organic matter (Attwood et al., 2014) and is highly acidic with pH ranging from 0.5-2 in summer and 1-3 in winter (Guo et al., 2015). Sulfate mainly forms from the oxidation of $SO_2$ that is primarily emitted from fossil fuel combustion (Wuebbles and Jain, 2001; Chin and Jacob, 1996). SOA accounts for a significant fraction of organic $PM_{2.5}$ in SE US (Lee et al., 2010) and is expected to derive primarily from isoprene (Ying et al., 2015). Together, high isoprene, sulfate and aerosol acidity make the SE US prime for the formation of sulfate-influenced biogenic SOA, including organosulfates. The Southern Oxidant and Aerosol Study (SOAS) which took place in 01 June–15 July of 2013 was focused on studying the SOA formation in the SE US and its impact on air quality and climate. The ground site discussed in this paper was situated in Centreville, AL, a rural, forested site which is mainly characterized by high isoprene emissions and to a lesser extent by other biogenic VOC such as monoterpenes, and is affected by anthropogenic pollutants from nearby cities such as Montgomery, Birmingham and Tuscaloosa (Hagerman et al., 1997), thus an ideal location for studying organosulfates derived from biogenic VOC.

The organosulfate contribution to $PM_{2.5}$ organic mass is estimated to have an upper limit of 5.0 - 9.3 % in the SE US (Tolocka and Turpin, 2012), suggesting that organosulfates contribute significantly to organic aerosol mass in this region. A limited, but growing number of organosulfates have been accurately quantified against authentic standards. The most abundant organosulfates to be previously quantified, during SOAS 2013, using authentic standards include 2-methyltetrol sulfate (Budisulistiorini et al., 2015; Rattanavaraha et al., 2016), 2-methylglyceric acid sulfate (Budisulistiorini et al., 2015; Rattanavaraha et al., 2016), glycolic acid sulfate (Liao et al., 2015; Hettiyadura et al., 2015; Rattanavaraha et al., 2016), lactic acid sulfate (Hettiyadura et al., 2015) and hydroxyacetone sulfate (Budisulistiorini et al., 2015; Hettiyadura et al., 2015). The quantification of organosulfates is currently limited by very few atmospherically relevant standards being commercially

available, requiring the development of standards by synthesis (Olson et al., 2011; Staudt et al., 2014; Hettiyadura et al., 2015; Rattanavaraha et al., 2016). In the absence of authentic standards, surrogate standards are commonly used, but can lead to significant and often uncharacterized biases that result from differences in negative electrospray ionization ((-) ESI) efficiencies (Staudt et al., 2014). Authentic standards are thus required for accurate quantification and determination of their contributions to PM mass.

Mechanistic studies have revealed pathways by which organosulfates form and have been reviewed elsewhere (Hallquist et al., 2009; Surratt et al., 2010; Ervens et al., 2011; Darer et al., 2011; Riva et al., 2015). Here, we focus on the most abundant organosulfates that have been quantified against authentic standards in the SE US during SOAS. 2-Methyltetrol sulfates primarily form by acid-catalyzed nucleophilic addition of sulfate to isoprene epoxydiols (IEPOX) (Surratt et al., 2010). They may also form by nucleophilic substitution of nitrate in organonitrates with sulfate (Darer et al., 2011) and isoprene ozonolysis in the presence of acidified sulfate seed aerosol (Riva et al., 2016). 2-Methylglyceric acid sulfate forms from either methacrylic acid epoxide (Lin et al., 2013) or hydroxymethyl-methyl-α-lactone (isoprene oxidation products), similarly to 2-methyltetrol sulfates, in the presence of sulfate under high $NO_x$ conditions (Nguyen et al., 2015). Glycolic acid sulfate forms more efficiently from glycolic acid relative to glyoxal in the presence of acidic sulfate (Liao et al., 2015), while both precursors have biogenic and anthropogenic origins they mainly form from isoprene oxidation in SE US (Liao et al., 2015; Fu et al., 2008). Lactic acid sulfate and hydroxyacetone sulfate can form via isoprene photo-oxidation in the presence of acidic sulfate (Surratt et al., 2008). Alternatively, lactic acid sulfate may derive from 2-E-pentenal, a photolysis product of 3-Z-hexenal (Shalamzari et al., 2016), while hydroxyacetone sulfate can form from isoprene ozonolysis in the presence of acidified sulfate seed aerosol (Riva et al., 2016). In addition, sulfate radical-induced oxidation can form organosulfates under acidic conditions; following this pathway, methyl vinyl ketone (MVK) can generate 2-methylglyceric acid sulfate, glycolic acid sulfate, lactic acid sulfate and hydroxyacetone sulfate while methacrolein (MACR) can form 2-methylglyceric acid sulfate and hydroxyacetone sulfate (Schindelka et al., 2013; Nozière et al., 2010). Organosulfate product distributions are thus expected to depend on precursor gas concentrations, acidity, and oxidant concentrations.

Mass spectrometry (MS) with (-) ESI is widely used to detect organosulfates (Iinuma et al., 2007; Gómez-González et al., 2008; Altieri et al., 2009; Reemtsma et al., 2006; Romero and Oehme, 2005). The bisulfate anion ($HSO_4^-$ at *m/z* 97) and sulfate radical anion ($SO_4^{-\bullet}$ at *m/z* 96) are characteristic fragment ions of organosulfates (Gómez-González et al., 2008; Romero and Oehme, 2005; Surratt et al., 2008). Tandem mass spectrometry ($MS^2$), in which precursors to these ions can be used as a means of identifying organosulfates and semi-quantifying them in the absence of authentic standards (Stone et al., 2009). The inherent limitations of this approach to semi-quantitation are discussed in Sect. 3.3. Offline MS detection of organosulfates is often coupled with liquid chromatography (LC). Reverse phase LC-MS methods are suitable for separation of aromatic and monoterpene derived organosulfates that contain hydrophobic moieties (e.g. aromatic rings or long alkyl chains) (Stone et al., 2012), but do not retain carboxy- and polyhydroxy-organosulfates that instead co-elute with sulfate and other organic compounds. Hydrophilic interaction liquid chromatography (HILIC) has been demonstrated to have

complementary selectivity to reversed phase separation and is preferred for retention of carboxyl-containing organosulfates (Hettiyadura et al., 2015).

Filter-based aerosol measurements are subject to sampling artifacts, such as gas adsorption on to quartz fiber filters (QFF) during sampling (Zhu et al., 2012; Turpin et al., 2000; Turpin et al., 1994). A recent study demonstrated formation of organosulfates from β-pinene oxide (a gas phase monoterpene oxidation product) adsorbed onto QFF, suggesting that organosulfates can also form on QFF during sampling (Kristensen et al., 2016). Thus, characterizing the extent of artifacts in ambient sampling is needed to ensure accurate measurements of organosulfates.

The central objectives of this study include; i) quantification of select organosulfates in $PM_{2.5}$ collected from Centreville, AL from 13 June – 13 July 2013 during SOAS against authentic standards, ii) assessment of correlations of select organosulfates with co-located measurements such as sulfate, aerosol water, aerosol acidity and potential VOC precursors, iii) evaluation of the extent of positive filter sampling artifacts associated with select organosulfates and iv) identification of the major organosulfates in Centreville, AL using semi-quantitative $MS^2$. For the latter two objectives a smaller subset of samples collected from 07 – 11 July 2013 were analyzed and the extent to which these samples represent the typical conditions observed during SOAS is discussed in Sect. 3.2 and 3.3. Through these efforts we expand the understanding of organosulfates in Centreville, AL during SOAS 2013 and constrain the extent to which positive filter sampling artifacts affect quantitation. In addition, the major organosulfates identified during this study provide new insights to the organosulfates that should be targets for future standard development.

## 2 Materials and methods

### 2.1 Chemicals and reagents

Methyl sulfate (sodium salt, 99 %, Acros Organics) and ethyl sulfate (sodium salt, 96.31 %, Sigma-Aldrich) standards were purchased. Benzyl sulfate (70.1 %) sodium salt was synthesized as described in Estillore et al. (2016). Hydroxyacetone sulfate and glycolic acid sulfate (potassium salts, > 95 %) were synthesized according to the method described in Hettiyadura et al. (2015). Lactic acid sulfate (24.9 %) was synthesized as described in Olson et al. (2011). Ultra-pure water was prepared on site (Thermo, Barnsted EasyPure-II; 18.2 MΩ cm resistivity, with OC < 40 µg/L). Other reagents included acetonitrile (Optima™, Fisher Scientific), ammonium acetate (≥ 99 %, Fluka, Sigma Aldrich) and ammonium hydroxide (Optima, Fisher Scientific).

### 2.2 $PM_{2.5}$ samples

$PM_{2.5}$ samples were collected from 13 June – 13 July, 2013 following the daytime (8:00-19:00 LT) and nighttime (20:00-7:00 LT) schedule. $PM_{2.5}$ was collected using two co-located medium-volume samplers (Teflon coated aluminum cyclone, URG-3000B, URG Corporation) on pre-baked (550 °C for 18 h) QFF (90 mm diameter, Pall Life Sciences). For the study of sampling artifacts during 07 - 11 July 2013, samplers were equipped with back-up QFF as described in section 2.3.

One field blank was collected for every five PM$_{2.5}$ samples following the same procedure, without passing air through the filters. All filter samples collected were stored in Al-foil (pre-baked at 550 °C for 5.5 h) lined petri dishes and were kept frozen (-20.0 °C) under dark conditions until extracted.

## 2.3 Positive filter sampling artifacts

5        Positive filter sampling artifacts associated with glycolic acid sulfate, lactic acid sulfate and hydroxyacetone sulfate from 07-11 July 2013 were assessed using filter samples collected on bare back-up QFF (Q$_B$) and sulfuric acid impregnated back-up QFF (Q$_{B-H_2SO_4}$; H$_2$SO$_4$ - 8.65 µg cm$^{-2}$) collected in series behind front QFF (Q$_F$) that collected PM$_{2.5}$ (Fig. S1). Q$_B$ were used to assess positive filter sampling artifacts due to gas adsorption on QFF, while Q$_{B-H_2SO_4}$ were used to assess positive filter sampling artifacts due to gas adsorption and reactions of gas phase precursors of organosulfates with sulfuric acid during

sample collection. Positive filter sampling artifacts (% $f_{artifacts}$) were calculated as the percent of organosulfate (X) on Q$_B$ or Q$_{B-H_2SO_4}$ relative to Q$_F$, according to Eq. (1):

$$\% f_{artifacts} = \left( \frac{[X\,(ng\,m^{-3})]_{back\_up\,filter}}{[X\,(ng\,m^{-3})]_{front\,filter}} \right) \times 100\% \tag{1}$$

## 2.4 Sample preparation

       Filter samples collected were prepared for the chemical analysis as described in Hettiyadura et al. (2015). Briefly,

portions of filters (~ 15 cm$^2$) were extracted by sonication (20 min, 60 sonics min$^{-1}$, 5510, Branson) with acetonitrile and ultra-pure water (95:5, 10 mL), filtered through polypropylene membrane syringe filters (0.45 µm pore size, Puradisc$^{TM}$25PP, Whatman$^®$), and reduced the volume to 500 µL under a stream of ultra-high purity nitrogen gas ($\leq$ 5 psi) at 50 °C using an evaporation system (Turbovap$^®$ LV, Caliper Life Sciences). Then the extracts were transferred to LC vials (1.5 mL, Agilent) and evaporated to dryness under a very light stream of ultra-high purity nitrogen gas at 50 °C using a microscale nitrogen

evaporation system (Reacti-Therm III TS 18824 and Reacti-Vap I 18825, Thermo Scientific) and then reconstituted in 300 µL acetonitrile: ultra-pure water (95:5).

## 2.5 Chemical analysis

### 2.5.1 Organic carbon (OC)

       OC mass was measured on 1.0 cm$^2$ punches of PM$_{2.5}$ sampled on Q$_F$ using a thermal-optical analyzer (Sunset

Laboratory, Forest Grove, OR, USA) according to the Aerosol Characterization Experiment (ACE)-Asia protocol described in Schauer et al. (2003).

### 2.5.2 Co-located measurements during SOAS

PM$_{2.5}$, sulfate, aerosol acidity, aerosol water, isoprene, glyoxal, formaldehyde, MACR, MVK, hydroxyacetone, glycolaldehyde, isoprene hydroxyl nitrates (ISOPN), isoprene hydroxyl hydroperoxides (ISOPOOH) and IEPOX measured during SOAS from 13 June – 13 July 2013 were averaged across sample collection time. The methods used for quantification of each of these were published elsewhere: PM$_{2.5}$ and sulfate (Edgerton et al., 2006), aerosol acidity and aerosol water (Guo et al., 2015) and VOC precursors (Kaiser et al., 2016); isoprene, MACR and MVK (Goldan et al., 2004; Gilman et al., 2010), glyoxal (Huisman et al., 2008), formaldehyde (Hottle et al., 2009; DiGangi et al., 2011), hydroxyacetone and ISOPN (Crounse et al., 2006), glycolaldehyde, ISOPOOH and IEPOX  (St. Clair et al., 2010).

### 2.5.3 Quantification of organosulfates using HILIC-TQD

Organosulfates were quantified using HILIC-TQD following Hettiyadura et al., (2015). Briefly, an ultra-performance liquid chromatography (UPLC; ACQUITY UPLC H-Class, Waters, Milford, MA, USA) with (-) ESI TQD (AQCUITY, Waters) was employed in multiple reaction monitoring (MRM) mode. Optimized MS conditions (cone voltages and collision energies) used for each analyte in MRM mode were given in Hettiyadura et al. (2015). Organosulfates were separated using HILIC on an ethylene bridged hybrid amide (BEH-amide) column (2.1 $\times$ 100 mm, 1.7 µm particle size; AQCUITY UPLC Waters) using an acetonitrile rich eluent with 10 mM ammonium acetate buffered to pH 9 by adjustment with ammonium hydroxide. The aqueous portion of the eluent was held at 5 % for two minutes, then increased to ~19 % over two minutes and held constant until 11 minutes before column re-equilibration. The instrument was calibrated daily with a freshly-prepared seven point calibration standard series (0.500-500. µg L$^{-1}$). Data were acquired and processed using MassLynx software (version 4.1). All measurements were field blank subtracted.

The analytical uncertainty in organosulfate concentrations were calculated from the total relative uncertainty (% e$_T$) propagated according to Eq. (2), accounting for the relative errors in air volume (% e$_V$, 5 %), extraction efficiency (% e$_E$; which represents the difference in the observed and expected responses of quality control samples to which  known amounts of analytes were added), and the relative error in instrumental analysis (% e$_I$; which is propagated from the instrument limit of detection and relative standard deviation of each organosulfate reported in Hettiyadura et al., 2015). For measurements requiring sample dilution, an additional error term (% e$_D$) was propagated considering the errors in initial and final volumes.

$$\% \, e_T = \sqrt{(\% \, e_V^2 + \% \, e_E^2 + \% \, e_I^2 + \% \, e_D^2)} \tag{2}$$

### 2.5.4 Qualitative analysis of major organosulfates in Centreville, AL

Major organosulfates in Centreville, AL were operationally defined as ions that fragmented to bisulfate anion (*m/z* 97) or sulfate radical anion (*m/z* 96) using HILIC-TQD in precursor ion mode across the  mass range 100-400 Da. For the precursors of *m/z* 97 a cone voltage of 28 V and a collision energy of 16 eV were used. For the precursors of *m/z* 96 a cone voltage of 42 V and a collision energy of 20 eV were used. The identified organosulfates underwent further characterization

using UPLC (ACQUITY UPLC, Waters; Milford, MA, USA) coupled with (-) ESI time-of-flight mass spectrometry (TOF-MS) (Bruker Daltonics MicrOTOF). HILIC separation was performed as described previously (Sect. 2.5.3.), with a different capillary voltage of 2.8 kV, a sampling cone voltage of 30 V and a desolvation gas flow rate of 600 L h$^{-1}$. Data were collected in the mass range 100–400 Da in full scan mode. A peptide, Val-Tyr-Val (*m/z* 378.2029, Sigma-Aldrich), was used for continuous MS mass calibration. Molecular formulas were assigned considering the presence of $C_{0-500}$, $H_{0-100}$, $N_{0-5}$, $O_{0-50}$, $S_{0-2}$, odd and even electron state, and a maximum error of 6 mDa.

## 3 Results and discussion

### 3.1 Quantification of organosulfates in Centreville, AL

The ambient concentrations of the organosulfates quantified against authentic standards in $PM_{2.5}$ collected from Centreville, AL from 13 June – 13 July 2013 are summarized in Table 1, and the three most abundant species are shown in Fig. 1 along with $PM_{2.5}$, OC and sulfate concentrations. Glycolic acid sulfate was the most abundant organosulfate followed by lactic acid sulfate, hydroxyacetone sulfate, methyl sulfate, ethyl sulfate (detected only in 18 samples) and benzyl sulfate (detected only in 1 sample).

Glycolic acid sulfate and lactic acid sulfate quantified in this study ranged from 0.5 – 52.5 ng m$^{-3}$ and 0.5 – 36.7 ng m$^{-3}$, respectively (Table 1). At the nearby Birmingham, AL site during SOAS, similar organosulfate concentrations were reported: glycolic acid sulfate averaged 26.2 ng m$^{-3}$ and had a maximum value of 75.2 ng m$^{-3}$ while lactic acid sulfate (quantified using propyl sulfate as the surrogate standard) averaged 2.7 ng m$^{-3}$ with a maximum value of 10.5 ng m$^{-3}$ (Rattanavaraha et al., 2016). These levels are significantly higher than the levels of glycolic acid sulfate and lactic acid sulfate reported previously in Bakersfield, CA (an urban site) from 16-18 June, 2010 at 4.5 – 5.4 ng m$^{-3}$ and 0.6 – 0.7 ng m$^{-3}$, respectively (Olson et al., 2011). These data indicate higher levels of these organosulfates in the SE compared to the Southwestern US (California) during summer, but are limited by few measurements of organosulfates in the literature.

The total contribution of the organosulfates quantified using authentic standards was less than 0.3 % of OC (Table 1). Meanwhile, the estimated upper bound contribution of organosulfates to organic matter (OM) is 5.0 – 9.3 % in the SE US (Tolocka and Turpin, 2012). Assuming OM/OC of 1.8 (Tolocka and Turpin, 2012), the calculated contribution of the organosulfates quantified in this study comprise 0.7 % of OM. Mean concentrations of 2-methyltetrol sulfates and 2-methylglyceric acid sulfate reported by Rattanavaraha et al. (2016) during SOAS 2013 in Centreville, AL were 207.1 ng m$^{-3}$ and 10.2 ng m$^{-3}$, respectively, which accounted for 3.7% and 0.2% of OM, for an average OC concentration of 3.07 µg m$^{-3}$ in Centreville during SOAS 2013 and OM/OC of 1.8. Together, the organosulfates quantified against authentic standards in Centreville accounts for 4.7 % of OM. Additional species that contribute significantly to MS$^2$ organosulfate signals are qualitatively and semi-quantitatively examined in Sect. 3.3.

Correlations of hydroxyacetone sulfate, lactic acid sulfate and glycolic acid sulfate with co-located gas and aerosol measurements were used to gain insights to their potential precursors and conditions conducive to their formation (Table 2).

VOC measurements used for correlation analysis include ISOPOOH and IEPOX that are isoprene low NOx oxidation products (Paulot et al., 2009; Krechmer et al., 2015), formaldehyde, MACR, MVK and ISOPN that are isoprene high NOx oxidation products (Kaiser et al., 2015; Marais et al., 2016; Spaulding et al., 2003), hydroxyacetone and glycolaldehyde that are further oxidation products of MACR and MVK, respectively (Galloway et al., 2011; Spaulding et al., 2003), and glyoxal that form

from both isoprene low and high NOx oxidation pathways (Marais et al., 2016). Organosulfates have longer lifetimes compared to isoprene and isoprene VOC precursors; glycolic acid sulfate and lactic acid sulfate reported to be stable for 21 days under highly acidic conditions (Olson et al., 2011), whereas above VOC have life times of several hours or less with respect to OH radicals which is their major sink (Paulot et al., 2009; Gaston et al., 2014; Lee et al., 2014; Montzka et al., 1993; Orlando et al., 1999; Lee et al., 1995). These differences in lifetimes confound correlation analysis, because their concentrations will vary

on different time scales. Due to the longer lifetimes of organosulfates, they can also be transported to the sampling site from long distances affecting correlations with VOC precursors that are short lived thus locally formed. In addition, Rivera-Rios et al. (2014) has shown that MACR, MVK and formaldehyde form as artifacts from decomposition of ISOPOOH on metal surfaces during sampling and instrument analysis. Thus, there is a negative bias in the measurements of ISOPOOH and a positive bias in MACR, MVK and formaldehyde which may also influence their correlations. The strong inter-correlations

observed for glycolic acid sulfate, lactic acid sulfate and hydroxyacetone sulfate suggest that they have common precursors and/or formation pathways. All three organosulfates correlated with both low and high NOx isoprene oxidation products. The relatively higher correlations of these organosulfates with formaldehyde, MACR and hydroxyacetone suggest that the high $NO_x$ pathway may play a larger role in their formation; however, this is not supported by lower correlations observed with ISOPN and MVK that are also high $NO_x$ isoprene oxidation products. Thus, further work is required to better understand the

VOC precursors and formation pathways of these organosulfates and will likely require organosulfates with higher time resolution.

All three species had moderate to strong correlations with sulfate, but not with liquid water content or acidity, suggesting that neither aerosol water nor aerosol acidity limit organosulfate formation. Similar correlations were reported at Centreville for isoprene derived SOA, and were attributed to variation of sulfate compared to consistently high aerosol acidity

and high relative humidity observed during SOAS 2013 (Xu et al., 2015). Further, these correlations are consistent across other SOAS ground sites (Rattanavaraha et al., 2016; Budisulistiorini et al., 2015) indicating that the association of isoprene derived SOA with sulfate is a regional characteristic. The correlations of organosulfates derived from isoprene and sulfate in the SE US, suggests that sulfate is a key factor that influences biogenic SOA formation.

**3.2 Positive filter sampling artifacts for select organosulfates**

The positive filter sampling artifacts associated with the three most abundant organosulfates quantified in Sect. 3.1 (glycolic acid sulfate, lactic acid sulfate and hydroxyacetone sulfate, respectively) were assessed from 07 – 11 July, 2013. This time period followed several days with rain, thus had slightly lower average PM$_{2.5}$ (5.24 ± 1.68 µg m$^{-3}$), OC (2.00 ± 0.67 µg m$^{-3}$), sulfate (1.26 ± 0.66 µg m$^{-3}$) and organosulfate concentrations relative to the average concentrations from 13 June – 13

July 2013: PM$_{2.5}$ (7.52 ± 3.41 µg m$^{-3}$), OC (3.07 ± 1.35 µg m$^{-3}$), sulfate (1.78 ± 0.81 µg m$^{-3}$). Within the studied subset of days, the 09 July daytime and nighttime, and 10 July daytime concentrations (Fig. 1) were similar to the average conditions observed during SOAS, and are considered to be most representative of the average conditions at Centreville during SOAS. The potential for these organosulfates in the gas phase to form positive sampling artifacts by adsorption onto QFF was assessed by parallel analysis of Q$_F$ and Q$_B$. Of the ten Q$_B$ analyzed to assess positive filter sampling artifacts due to gas adsorption, only three contained detectable levels of glycolic acid sulfate and one contained detectable levels of lactic acid sulfate (Table 3). Maxima occurred in the 09 July nighttime sample with the backup filter containing 2.2 ± 0.8 % (0.30 ± 0.06 ng m$^{-3}$) of the PM$_{2.5}$ glycolic acid sulfate concentration and 1.1 ± 0.5 % (0.15 ± 0.07 ng m$^{-3}$) of the PM$_{2.5}$ lactic acid sulfate concentration. Meanwhile, hydroxyacetone sulfate was below the instrument detection limit in all Q$_B$ analyzed, and an upper limit of the positive artifact was estimated as 3.2 %. The positive filter sampling artifacts associated with these three organosulfates from gas adsorption were only detected sporadically and at very low levels (~ 3 %) that fell within the propagated analytical uncertainty and were considered to be negligible. These results are consistent with those of Kristensen et al. (2016) who did not observe these three organosulfates in the denuder samples collected upstream of Teflon filters on which these organosulfates were detected. Because the greatest sampling artifacts were observed on 09 July when PM$_{2.5}$ and OC loadings were greatest (Fig. 1), the extent of artifacts may be even greater on days with higher PM$_{2.5}$ and OC loadings.

The potential for glycolic acid sulfate, lactic acid sulfate and hydroxyacetone sulfate to form on QFF by acid catalyzed heterogeneous reactions was assessed by the parallel analysis of Q$_F$ with Q$_{B-H_2SO_4}$, in which the Q$_{B-H_2SO_4}$ filters were loaded with approximately twice the amount of sulfate that was expected to be collected on 90 mm QFF (with a total sampling area of 50.3 cm$^2$) over 11 hours of sampling at a flow rate of 92 lpm, based on an average PM$_{2.5}$ sulfate concentration of 4.11 ± 0.55 µg m$^{-3}$ in Centreville, AL (Edgerton et al., 2005). The organosulfates detected on Q$_{B-H_2SO_4}$ (maximum concentration, % $f_{artifacts}$) was highest for glycolic acid sulfate (0.8 ± 0.2 ng m$^{-3}$, 5.7 ± 2.1 %), then lactic acid sulfate (0.43 ± 0.08 ng m$^{-3}$, 3.7 ± 0.8 %) followed by hydroxyacetone sulfate (0.18 ± 0.05 ng m$^{-3}$, 4.7 ± 1.2 %). Concentrations of organosulfates formed on the Q$_{B-H_2SO_4}$ filters followed the same trend as their PM$_{2.5}$ concentrations (section 3.1), while the % $f_{artifacts}$ was relatively consistent across the detected organosulfates. Organosulfates were more frequently detected on the Q$_{B-H_2SO_4}$ compared to the Q$_B$ and at higher concentrations (Table 3), indicating that in addition to adsorption of organosulfates in the gas phase, organosulfate formation may occur on QFF by adsorption and reaction of gas-phase precursors of organosulfates with H$_2$SO$_4$. The maximum extent of the sulfuric acid-enhanced artifact formation was 4.5 – 7.8 % (Table 3), which is greater than gas adsorption alone, but is overall relatively low. Because the positive filter sampling artifacts were detected sporadically and only accounted for a minor fraction of the total organosulfate concentration that fell within the analytical uncertainty, the PM$_{2.5}$ organosulfate concentrations reported in section 3.1 were not corrected for positive filter sampling artifacts.

The extent of on-filter reactions to form glycolic acid sulfate, lactic acid sulfate, and hydroxyacetone sulfate appears to be site-specific. In a prior study in Hyytiälä, Finland, Kristensen et al. (2016) attributed the majority of organosulfates detected on high-volume filter samples to on-filter oxidation and sulfation reactions, including *m/z* corresponding to glycolic acid sulfate, lactic acid sulfate, and hydroxyacetone sulfate. However, for samples collected in Copenhagen, only 5 % of the

daytime average concentrations and 14 % of the nighttime average concentrations of the glycolic acid sulfate was attributed to on-filter reactions, similar to this study, while lactic acid sulfate and hydroxyacetone sulfate were reported to have negative sampling artifacts (Kristensen et al., 2016), which could result from degradation during sampling, sample preparation, or analysis. With varying extents of organosulfate sampling artifacts reported across sampling sites, it is recommended that sampling artifacts be evaluated at future field study sites.

### 3.3 Major organosulfates in Centreville, AL

A semi-quantitative analysis was conducted to identify the organosulfate species strongest $MS^2$ signals. Organosulfates were evaluated as precursor ions of the bisulfate anion ($HSO_4^-$ at $m/z$ 97) and sulfate radical anion ($SO_4^{-\bullet}$ at $m/z$ 96) that are characteristic of this group of compounds (Gómez-González et al., 2008; Riva et al., 2016; Surratt et al., 2008). This analysis was applied to samples collected from 07 – 11 July, 2013, with a focus on the 10 July daytime sample with levels of $PM_{2.5}$ (7.01 ± 0.80 µg m$^{-3}$), OC (2.63 ± 0.21 µg m$^{-3}$), sulfate (1.06 ± 0.17 µg m$^{-3}$) and organosulfates (Fig. 1) near to the study average (Sect. 3.2 and Table 1).

The ability of an organosulfate to contribute to the bisulfate ion signal depends on its individual (-) ESI ionization efficiency, $MS^2$ fragmentation patterns, and mass concentration. Absolute quantitation requires instrument calibration as discussed in Sect. 1; however, this is not possible for the vast majority of atmospheric organosulfates, because standards are not commercially available. In the following data analysis, it is assumed that organosulfates have an equal ability to form the bisulfate anion/sulfate radical anion, so that semi-quantitative insights may be gained to their relative abundance in ambient aerosol. This approach is limited by the fact that differing ionization efficiencies and fragmentation patterns have not been controlled and may introduce positive or negative biases. Consequently, the ranking in Table 4 should not be considered as an accurate measure of relative abundance, but a best estimate in the absence of authentic standards.

The limitations of this approach is illustrated by the comparison of the semi-quantitative behavior of glycolic acid sulfate, lactic acid sulfate and hydroxyacetone sulfate in their formation of the bisulfate anion and their absolute quantitation. For the 10 July 2013 daytime sample, the relative contribution to bisulfate ion signal was highest for hydroxyacetone sulfate (1.10 %), then glycolic acid sulfate (0.57 %) and lactic acid sulfate (0.23 %) respectively, while absolute concentrations of glycolic acid sulfate (14 ± 5 ng m$^{-3}$) and lactic acid sulfate (13 ± 2 ng m$^{-3}$) were greater than hydroxyacetone sulfate (8.5 ± 0.3 ng m$^{-3}$) (Fig. 1). The negative bias in the bisulfate ion signal towards later-eluting organosulfates results from the mobile phase gradient used in UPLC; when hydroxyacetone sulfate elutes ($t_R$ 0.69 min), the mobile phase is 95: 5 % acetonitrile and water compared to an average of 81: 19 % acetonitrile and water when glycolic acid sulfate ($t_R$ 7.82 min) and lactic acid sulfate ($t_R$ 7.54 min) elute. Acetonitrile has a higher vapor pressure than water and more readily desolvates in the ionization source. Thus, when increasing the water content of the eluent, the signal of later eluting ions decreases. Consequently, organosulfates retained longer on the BEH-amide column during HILIC gradient separation, such as organosulfates containing carboxyl and multiple hydroxyl groups are expected to be under-represented in this semi-quantitative analysis. These results emphasize the importance of using authentic standards to calibrate the instrument, particularly when using gradient elution. Nonetheless, it

is a valuable endeavor to gain semi-quantitative information on major organosulfate signals in order to guide future developments of authentic standards that will ultimately enables absolute quantitation.

A mass spectrum of the precursor ions to $m/z$ 97 integrated over the entire HILIC separation (0-11 min) for the 10 July daytime sample with the ten strongest signals marked is shown in Fig. 2. Each nominal $m/z$ in Fig. 2 corresponded to a single monoisotopic mass as determined from HILIC-TOF, except for $m/z$ 155 and 199. Table 4 ranks these ten organosulfate signals in order of decreasing relative contribution to the total bisulfate ion signal and summarizes their $m/z$, molecular formulae determined from HILIC-TOF, expected precursor(s) based on prior field and SOA chamber studies, and proposed molecular structures with consideration of results from prior studies, double bond equivalences, functional groups, and HILIC retention time. The 10 July nighttime (Fig. S2) and other daytime and nighttime samples collected from 07-11 July 2013 (Fig. S3) were analyzed in an analogous way. Because some organosulfates do not fragment to $m/z$ 97, precursors of the $m/z$ 96 was analyzed (Fig. S4, Table S1), although its signal was only 2% of the MS$^2$ response of $m/z$ 97.

The strongest organosulfate signals were associated with isoprene and its oxidation products. The dominant organosulfate signal was $C_5H_{11}SO_7^-$ (215.0225; Fig. 3a) that corresponded to 2-methyltetrol sulfates that predominantly form by the acid catalyzed nucleophilic addition of sulfate to IEPOX (Surratt et al., 2010). This species accounted for 42-62% of the bisulfate anion signal across the samples analyzed semi-quantitatively. Other major organosulfate signals that were consistently observed ($\geq$90% of the 10 samples) included $m/z$ 213, 211, 183 and 153 that have been associated with isoprene. The species with formula $C_5H_9SO_7^-$ (213.0069; Fig. 3b) has been observed to form from isoprene photo-oxidation in the presence of acidic sulfate under low NO$_x$ conditions (Surratt et al., 2008) and ozonolysis (Riva et al., 2016). Structurally, $C_5H_9SO_7^-$ is closely related to 2-methyltetrol sulfate, with one increasing unit of unsaturation. Likewise, $C_5H_7SO_7^-$ (210.9912; Fig 3c) is related to 2-methyltetrol sulfate by two units of unsaturation. An organosulfate with this formula has been observed in an isoprene chamber experiment, but may have other VOC precursors (Surratt et al., 2008). The short retention times (< 3 min) of $C_5H_9SO_7^-$ and $C_5H_7SO_7^-$ indicate the absence of carboxyl groups. These two organosulfates have been proposed to result from further oxidation of 2-methyltetrol sulfate followed by subsequent ring closing (Hettiyadura et al. 2015), although this has not been confirmed. The species $C_4H_7SO_6^-$ (182.9963) has multiple constitutional isomers (Fig. 3e) with the dominant peak eluting at 0.91 minutes. The product ion spectrum of $C_4H_7SO_6^-$ (Fig. S5) included signals (by chemical formula, observed mass, and error in mDa) at $HSO_3^-$ (80.9642, -0.4), $HSO_4^-$ (96.9593, -0.3), $C_3H_5SO_5^-$ (152.9856, -0.2) and $C_4H_5SO_5^-$ (164.9859, 0.1), thus corresponding to hydroxybutan-3-one-2-sulfate (Shalamzari et al., 2013) that is derived from isoprene oxidation products, MVK and MACR (Schindelka et al., 2013). Also among the strongest signals are $C_3H_5SO_5^-$ (152.9858; Fig. 3g) with the dominant isomer corresponding to hydroxyacetone sulfate (discussed in Sect. 3.1) and $C_5H_{11}SO_6^-$ (199.0276; Fig. 3h) an isoprene ozonolysis product (Riva et al., 2016) that can also form from MBO in the presence of oxidants and sulfate under low NO$_x$ conditions (Zhang et al., 2012a). In addition to $C_5H_{11}SO_6^-$, another organosulfate at nominal $m/z$ 199 corresponding to $C_4H_7SO_7^-$ (198.9912; Fig. S6) was identified by its product ion spectrum match to 2-methylglyceric acid sulfate (Fig S7 and Sect. 1). At the nominal $m/z$ 155 was glycolic acid sulfate discussed in Sect. 3.1 ($C_2H_3SO_6^-$; Fig. S8). A nitro-oxy organosulfate, $C_5H_{10}NSO_9^-$ (260.0076; Fig. S4 and Table S1) contributed up to 5.4 %

of the $m/z$ 96 precursor ion signal and is also associated with isoprene (Surratt et al., 2008; Gómez-González et al., 2008). The importance of these isoprene-derived organosulfates is also supported by their high abundance reported previously during SOAS 2013 at Look Rock, TN (Budisulistiorini et al., 2015), Birmingham, AL and Centreville (Rattanavaraha et al., 2016) during SOAS 2013. Together, these data demonstrate that isoprene chemistry dominates the formation of organosulfates in Centreville.

Organosulfates with formulas $C_7H_{11}SO_7^-$ (239.0225; Fig. 3d) and $C_{10}H_{16}NSO_{10}^-$ (342.0495; Fig. 3f) were also among the strongest signals identified from precursors of $m/z$ 97 scan and have been associated with monoterpene SOA formed in the presence of acidic sulfate (Surratt et al., 2008). Other monoterpene derived organosulfates identified from bisulfate ion signal include $C_{10}H_{17}SO_7^-$ (281.0695; observed in 90 % of the samples analyzed) $C_{10}H_{17}SO_8^-$ (297.0644), $C_7H_{11}SO_6^-$ (223.0276) and $C_{10}H_{15}SO_7^-$ (279.0538) (Table S2 and Fig. S2 and S3). Monoterpene-derived nitro-oxy organosulfates were particularly responsive to precursors of $m/z$ 96; $C_{10}H_{16}NSO_{10}^-$ (342.0495), $C_{10}H_{16}NSO_8^-$ (310.0597) and $C_{10}H_{16}NSO_7^-$ (294.0647) (Table S1 and Fig. S4). The nitro-oxy organosulfate $C_{10}H_{16}NSO_7^-$ (294.0647) accounted for 25 % of the total $m/z$ 96 signal in $PM_{2.5}$ sample collected during nighttime on 10 July 2013 (Table S1). This semi-quantitative result is consistent with prior field studies reported $m/z$ 294 as the most abundant nitro-oxy organosulfate in SE US, particularly during night time (Gao et al., 2006; Surratt et al., 2008).

The major organosulfate signal at $m/z$ 155 in the bisulfate ion scan corresponded to $C_3H_7SO_5^-$ (155.0014; Fig. 3j and Fig. S8) and was not previously reported. Its molecular formula and double bond equivalence suggest that the non-sulfate oxygen is likely a hydroxyl group (Table 4). Other major organosulfate signals identified from $m/z$ 96 that were not previously reported in the atmosphere include $C_4H_7SO_4^-$ (151.0065), $C_3H_5SO_4^-$ (136.9909) and $C_5H_8NO_8S^-$ (241.9971) (Table S1). Based on the molecular formula and double bond equivalence (Table S1), $m/z$ 151 is suggested as a methylallyl sulfate, $m/z$ 137 may be allyl sulfate and $m/z$ 242 may be a nitro-oxy organosulfate with a carbonyl group. However, the precursors to these organosulfates are unknown.

The organosulfate with the formula $C_{12}H_{25}SO_4^-$ (265.1474; Fig. 3i) is consistent with dodecyl sulfate (a.k.a. lauryl sulfate), the most common surfactant use in manufacture of cleaning and hygiene products. A single peak with a very short retention time is consistent with a largely aliphatic structure. Anionic surfactants including dodecyl sulfate have been observed in aerosol generated from waste water (Radke, 2005) and in coastal sea spray aerosol (Cochran et al., 2016). While sea spray was observed to impact the Centreville site on some days during SOAS (Allen et al., 2015), it was not a major source on the dates discussed herein, pointing towards waste water as a possible origin.

Together, the ten highest organosulfate signals in each sample analyzed (Fig. 2, S2 and S3) contributed 58-78 % of the total bisulfate ion signal, with the tenth greatest intensity signal accounting for 0.25 to 1.12 % of the total bisulfate ion signal. From the remaining organosulfate signal, we estimate a minimum of ~20-200 other minor organosulfates are present in Centreville, AL. In summary, a few highly abundant organosulfate species (e.g. 2-methyltetrol sulfates) dominate the bisulfate ion signal, while a relatively large number of minor organosulfate species are present in Centreville during the summer.

The semi-quantitative results of organosulfates are both consistent and complementary to Riva et al. (2016) during SOAS. Five of the thirteen organosulfates quantified by Riva et al. (2016) in Centreville were among the ten major organosulfate signals observed herein; these included isoprene photo-oxidation products $C_5H_{11}SO_7^-$ (215.0225), $C_5H_9SO_7^-$ (213.0069), $C_3H_5SO_5^-$ (152.9858) and isoprene ozonolysis products $C_4H_7SO_6^-$ (182.9963) and

$C_5H_{11}SO_6^-$ (199.0276). Other organosulfates, with $m/z$ 181, 201, 227, 249, 267 and 315 were reported to have lower relative abundance (Riva et al., 2016) and were not among the ten major organosulfates in this study. Meanwhile, the organosulfate with $m/z$ 197 ($C_5H_9SO_6^-$) was reported to be relatively high in Centreville (Riva et al., 2016), but was not identified as a major organosulfate in our study, likely due to differences in semi-quantitation methods. Together, these data demonstrate that organosulfates in Centreville are primarily derived from isoprene. In addition, our semi-quantiative analysis demonstrates

relatively strong organosulfate signals from monoterpenes and to a lesser extent anthropogenic sources at Centreville.

### 3.4 Tentative identification of 2-methyltetrol sulfate isomers

HILIC chromatography resolved six, baseline resolved peaks of 2-methyltetrol sulfates ($C_5H_{11}SO_7^-$; Fig. 3a) with retention times consistent with those reported by Hettiyadura et al. (2015). Based on the structures of β- and δ-IEPOX (Paulot et al., 2009), it is possible that the resulting 2-methyltetrol sulfate include the sulfate moiety at primary, secondary or tertiary

positions. The position of the sulfate group in 2-methyltetrol sulfates were tentatively identified by their relative acid hydrolysis rates as primary (most stable), secondary (intermediate stability), or tertiary (least stable; as discussed in the SI and shown in Fig. S9). These assignments are based upon their enthalpy of hydrolysis and neutral hydrolysis lifetime reported by Darer et al. (2011) and Hu et al. (2011). Accordingly, the first two 2-methyltetrol sulfate peaks to elute were assigned as diastereomers of the tertiary conformation, the middle two peaks as diastereomers of the secondary conformation, and the last two peaks as

diastereomers of primary 2-methyltetrol sulfate (Fig. 3a and Fig. S9). The relative contribution of these peaks to the bisulfate anion signal in order of elution were 23.9 %, 10.5 %, 23.4 %, 41.0 %, 0.8 %, and 0.4 % (Table 4). With a negative bias in peak area for late-eluting peaks, these percentages are expected to underestimate the contribution from primary organosulfates. With this knowledge, we expect that 2-methyltetrol sulfates have appreciable contributions from primary, secondary, and tertiary organosulfates. Confirmation of the configuration and their absolute quantitation would be made possible through synthesized

standards.

### 4 Conclusions

The three most abundant organosulfates quantified using authentic standards in $PM_{2.5}$ collected from Centreville, AL from 13 June – 13 July, 2013 were glycolic acid sulfate, lactic acid sulfate and hydroxyacetone sulfate respectively. Their ambient concentrations correlated with sulfate and biogenic VOC precursors, particularly isoprene and its oxidation products,

indicating their importance in organosulfate formation in the SE US. Positive filter sampling artifacts associated with these three organosulfates due to gas adsorption were negligible ($\%f < 3\,\%$), while sulfuric acid enhanced the positive filter sampling

artifacts, but were relatively small (%$f$ < 7.8 %). Thus the organosulfates quantified using $PM_{2.5}$ sampled on QFF in Centreville, AL during SOAS 2013 considered to have negligible to minor positive filter sampling artifacts.

The precursor ion scan of the bisulfate anion ($m/z$ 97) and sulfate radical anion ($m/z$ 96) were used semi-quantitatively to identify major organosulfate species in ambient aerosol in the Centreville, AL. From the ten strongest responding ions identified, 2-methyltetrol sulfates accounted for nearly half of the total bisulfate ion signal in all samples analyzed. By comparison to chamber studies, the major organosulfates identified in this study derive mainly from biogenic VOC, mainly isoprene, and to a lesser extent monoterpenes and MBO. Five of the ten major organosulfates identified from the bisulfate ion signal are consistent with those reported by Riva et al. (2016) in Centreville during SOAS and thus reinforce their conclusions that ozonolysis and photochemical reactions of isoprene influence the organosulfate levels and composition in Centreville. The organosulfates in Centreville, AL were dominated by few major species among large number of minor species. Even in areas heavily influenced by biogenic SOA, like Centreville, organosulfates may have primary sources, such as dodecyl sulfate ($C_4H_7SO_6^-$; 265.1474) that is expected to originate from wastewater.

The precursors of bisulfate anion and sulfate radical anion provide insights for the major organosulfate species in SE US that should be targets for future organosulfate standard development:

i. 2-Methyltetrol sulfates in Centreville have a sizable contribution from primary, secondary and tertiary isomers. Because of their different atmospheric lifetimes (e.g., towards hydrolysis (Darer et al., 2011; Hu et al., 2011)), the relative amounts of these isomers may provide insights to the ageing and fate of anthropogenically influenced isoprene-derived SOA. To facilitate this, future studies should focus on synthesizing standards and quantifying each of these isomers.

ii. The isoprene related organosulfates, $C_5H_9SO_7^-$ (213.0069) and $C_5H_7SO_7^-$ (210.9912) contributed ~ 4 % each of the total bisulfate ion signal (Table 4), which suggest that they are relatively abundant in Centreville and prime targets for standard development. Further, the lower retention times of these two organosulfates on BEH-amide column during HILIC separation reflects an absence of carboxylic acid groups and point towards to the structures proposed by Hettiyadura et al. (2015).

iii. Multiple isomers of many organosulfates are observed with HILIC chromatography that co-elute under reversed-phase LC conditions. HILIC-MS/MS provides a basis for assessing the relative abundance of isomers and indicate that 1-hydroxybutane-3-one-2-sulfate is the dominant isomer of $C_4H_7SO_6^-$ (182.9963) in Centreville, AL. Likewise, $C_{10}H_{16}NSO_{10}^-$ (342.0495) and $C_7H_{11}SO_7^-$ (239.0225) are expected to be among abundant monoterpene derived organosulfates in Centreville, AL. Similar to Riva et al. (2016), $C_5H_{11}SO_6^-$ (199.0276) is relatively abundant in Centreville, but further experiments are need to identify its origin. Because of their relatively strong MS[2] signals, these species are also strong candidates for standard development and/or quantification in ambient aerosol.

Future efforts at standard development should focus on organosulfates that are expected to have high abundance, frequently detected in ambient aerosol, and/or have high specificity to VOC precursors.

## 5 Data availability

The SOAS research data used in this publication are available at http://esrl.noaa.gov/csd/groups/csd7/measurements/2013senex/Ground/DataDownload/.

## Acknowledgements

5        The authors would like to thank E. Geddes, K. Richards, and T. Humphrey at Truman State University for synthesizing standards of benzyl sulfate, hydroxyacetone sulfate, and glycolic acid sulfate; S. Staudt at University of Wisconsin, Madison for synthesizing the lactic acid sulfate standard; L. Teesch and V. Parcell for their assistance in University of Iowa High Resolution Mass Spectrometry Facility (HRMSF); Ruikun Xin for his assistance in data processing; Eric Edgerton at Atmospheric Research & Analysis for providing $PM_{2.5}$ and sulfate measured in Centreville, AL during SOAS 2013; R. Weber, H. Guo and A. Nenes at Georgia Institute of Technology for providing aerosol acidity and aerosol water measured during SOAS 2013; P. Wennberg, T. Nguyen, J. St. Clair at California Institute of Technology for access to glycolaldehyde, hydroxyacetone, IEPOX, ISPOOH and ISOPN measurements; and A. Carlton from Rutgers University and J. Jimenez from University of Colorado, Boulder for organizing the SOAS component of Southeast Atmosphere Study. We would also like to thank US EPA Science To Achieve Results (STAR) program (grant number 83540101) for funding this research.

## Disclaimer

The content of this article is solely the response of the authors and do not necessarily represent the official views of the US EPA. Further, US EPA does not endorse the purchase of any commercial products or services mentioned in the publication.

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

Figure 1: Concentrations of glycolic acid sulfate, lactic acid sulfate and hydroxyacetone sulfate quantified against authentic standards, and PM$_{2.5}$, organic carbon (OC) and sulfate (SO$_4^{2-}$) measured at Centreville, AL during SOAS from 13 June – 13 July 2013 on the basis of day (D; 08:00-19:00 LT) and night (N; 20:00-07:00 LT). Error bars represent the analytical uncertainty.

Figure 2: The mass spectrum of precursors of *m/z* 97 (HSO$_4^-$) for the daytime sample collected on 10 July 2013 using HILIC-TQD. The ten labeled *m/z* correspond to the deprotonated ions ([M-H]$^-$) of organosulfates with the greatest intensity.

Figure 3: Extracted chromatograms for the ten major organosulfates (monoisotopic mass ± 0.01 Da) in the daytime sample collected on 10 July 2013 using HILIC-TOF.

**Table 1**: Concentration (range and mean ± one standard deviation) of each organosulfate quantified against authentic standards from 13 June – 13 July 2013 in Centreville, AL and their mean contribution (%) to $PM_{2.5}$ and OC (± one standard deviation).

| Organosulfate | Concentration (ng m$^{-3}$) | | Mean contribution (%) | |
| --- | --- | --- | --- | --- |
| | Range | Mean | $PM_{2.5}$ | OC |
| glycolic acid sulfate | 0.5 – 52.5 | 20.6 ± 14.3 | 0.2 ± 0.1 | 0.09 ± 0.04 |
| lactic acid sulfate | 0.5 – 36.7 | 16.5 ± 10.3 | 0.2 ± 0.1 | 0.10 ± 0.05 |
| hydroxyacetone sulfate | 0.5 – 14.3 | 5.8 ± 3.1 | 0.08 ± 0.03 | 0.04 ± 0.02 |
| methyl sulfate | 0.2 – 9.3 | 1.8 ± 2.4 | 0.03 ± 0.04 | 0.007 ± 0.008 |

**Table 2**: Correlations of hydroxyacetone sulfate, lactic acid sulfate and glycolic acid sulfate with each other and co-located measurements: isoprene, glyoxal that is form from both isoprene high and low NOx oxidation pathways, formaldehyde, isoprene hydroxyl nitrates (ISOPN), methacrolein (MACR), methylvinyl ketone (MVK) that are isoprene high NOx oxidation products, hydroxyacetone and glycolaldehyde that are further oxidation products of MACR and MVK, respectively, and isoprene hydroxyl hydroperoxides (ISOPOOH) and isoprene dihydroxy epoxides (IEPOX) that are isoprene low NOx oxidation products, sulfate, aerosol water and aerosol acidity. Underlined correlation coefficients are statistically significant at 95 % confidence interval.

| Co-located measurement | Number of samples | Pearson correlation coefficient (r) | | |
| --- | --- | --- | --- | --- |
| | | Glycolic acid sulfate | Lactic acid sulfate | Hydroxyacetone sulfate |
| Lactic acid sulfate | 60 | | | 0.86 |
| Glycolic acid sulfate | 60 | | 0.88 | 0.71 |
| Isoprene | 59 | 0.44 | 0.40 | 0.45 |
| Glyoxal | 60 | 0.60 | 0.65 | 0.56 |
| Formaldehyde | 60 | 0.73 | 0.76 | 0.69 |
| ISOPN | 42 | 0.32 | 0.40 | 0.30 |
| MACR | 59 | 0.67 | 0.67 | 0.59 |
| MVK | 59 | 0.30 | 0.43 | 0.35 |
| Hydroxyacetone | 42 | 0.68 | 0.70 | 0.63 |
| Glycolaldehyde | 38 | 0.43 | 0.50 | 0.39 |
| ISOPOOH | 38 | 0.52 | 0.48 | 0.32 |
| IEPOX | 38 | 0.40 | 0.41 | 0.14 |
| Sulfate | 60 | 0.69 | 0.74 | 0.63 |
| Aerosol water | 56 | 0.32 | 0.26 | 0.33 |
| Aerosol acidity ([H$^+$]) | 49 | -0.14 | 0.13 | 0.20 |

**Table 3**: Positive filter sampling artifacts associated with the three most abundant organosulfates quantified in Centreville, AL from 07-11 July 2013 due to gas adsorption alone and gas adsorption and reaction of VOC precursors of organosulfates with sulfuric acid. Given in the table are frequency of detection (FOD, n = 10) and positive filter sampling artifacts as a fraction of their PM$_{2.5}$ concentrations (%$f$).

| Organosulfate | Artifacts by gas adsorption | | Artifacts by gas adsorption and reaction of VOC precursors of organosulfates with sulfuric acid | |
|---|---|---|---|---|
| | FOD (%) | %$f$ artifacts (max) | FOD (%) | %$f$ artifacts (max) |
| glycolic acid sulfate | 30 | $2.2 \pm 0.8$ | 60 | $5.7 \pm 2.1$ |
| lactic acid sulfate | 10 | $1.1 \pm 0.5$ | 20 | $3.7 \pm 0.8$ |
| hydroxyacetone sulfate | 0 | - | 40 | $4.7 \pm 1.2$ |

**Table 4:** The ten organosulfates with the strongest contributions to the bisulfate product ion signal in Centreville, AL for the daytime sample collected on 10 July 2013. The ten organosulfates were ranked in the order of the greatest contribution to the bisulfate product ion signal. Summarized for each signal are formula determined using high resolution ToF MS, the calculated monoisotopic mass ([M-H]⁻), proposed structure (with reference to the article proposing the structure), VOC precursor(s) indicated by SOA chamber studies, retention time(s) ($t_R$) on the BEH-amide column during HILIC gradient separation (solvent peak at 0.38 min), error in $m/z$ (mDa) for each peak, and the relative contribution of each peak and the total peak area to the total bisulfate product ion signal. Many organosulfates are likely to have multiple isomers, although only one isomer is shown.

| Rank | [M-H]⁻ | | Structure | VOC precursor(s) | $t_R$ (min) | Error (mDa) | Contribution to total bisulfate signal (%) | |
|---|---|---|---|---|---|---|---|---|
| | Formula | Mass | | | | | by peak | total |
| 1 | $C_5H_{11}SO_7^-$ | 215.0225 |  (Surratt et al., 2010) | Isoprene (Surratt et al., 2007a; Surratt et al., 2007b; Surratt et al., 2010) | 1.40 1.74 2.87 3.65 4.49 4.83 | -0.6 0.5 -0.3 -1.7 -0.2 0.1 | 10.35 4.53 10.12 17.75 0.35 0.17 | 43.27 |
| 2 | $C_5H_9SO_7^-$ | 213.0069 |  (Hettiyadura et al., 2015) | Isoprene (Surratt et al., 2008) | 1.10 1.29 1.4 – 1.65 1.80 1.9 – 2.8 | -0.4 0.5 1.2 0.7 1.4 | 0.47 0.17 0.22 0.62 3.44 | 4.91 |
| 3 | $C_5H_7SO_7^-$ | 210.9912 |  (Hettiyadura et al., 2015) | Isoprene (Surratt et al., 2008) | 0.56 0.67 0.74 0.85 | -1.5 -1.4 0.9 -1.4 | 0.42 1.63 0.63 1.59 | 4.27 |
| 4 | $C_7H_{11}SO_7^-$ | 239.0225 |  (Nozière et al., 2010) | Limonene (Surratt et al., 2008) | 0.58 0.67 0.74 0.80 0.91 1.00 | 4.5 -0.4 -1.4 0.5 -0.8 -1.2 | 0.05 0.33 0.36 0.25 0.16 0.65 | 1.81 |
| 5 | $C_4H_7SO_6^-$ | 182.9963 |  (Shalamzari et al., 2013) | Isoprene (Riva et al., 2016) Methyl vinyl ketone and methacrolein (Schindelka et al., 2013) | 0.67 0.83 0.91 1.00 1.23 | 1.2 1.0 -0.6 -1.8 -0.1 | 0.05 0.20 1.02 0.22 0.25 | 1.73 |

**Table 4: (Cont.)**

| Rank | [M-H]⁻ | | Structure | VOC precursor(s) | $t_R$ (min) | Error (mDa) | Contribution to total bisulfate signal (%) | |
|---|---|---|---|---|---|---|---|---|
| | Formula | Mass | | | | | by peak | total |
| 6 | $C_{10}H_{16}NSO_{10}^-$ | 342.0495 |  (Yassine et al., 2012, supporting information) | α-terpinene and α and β-pinene (Surratt et al., 2008) | 0.54 0.61 | 0.3 0.7 | 1.13 0.36 | 1.49 |
| 7 | $C_3H_5SO_5^-$ | 152.9858 |  (Surratt et al., 2010) | Isoprene (Surratt et al., 2008) | 0.69[a] 0.91 4.34 | -1.1 -1.2 -0.7 | 1.10 0.12 0.05 | 1.27 |
| 8 | $C_5H_{11}SO_6^-$ | 199.0276 |  (Zhang et al., 2012b) | MBO (Nozière et al., 2010) Isoprene (Riva et al., 2016) | 1.05 1.89 | -0.9 0.8 | 0.46 0.57 | 1.03 |
| 9 | $C_{12}H_{25}SO_4^-$ | 265.1474 |  | Anthropogenic | 0.54 | 0.0 | 0.98 | 0.98 |
| 10 | $C_3H_7SO_5^-$ | 155.0014 |  | Unknown | 1.23 1.31 | -0.5 -0.7 | 0.32 0.30 | 0.62 |

[a]structure was confirmed using a synthesized authentic standard of hydroxyacetone sulfate

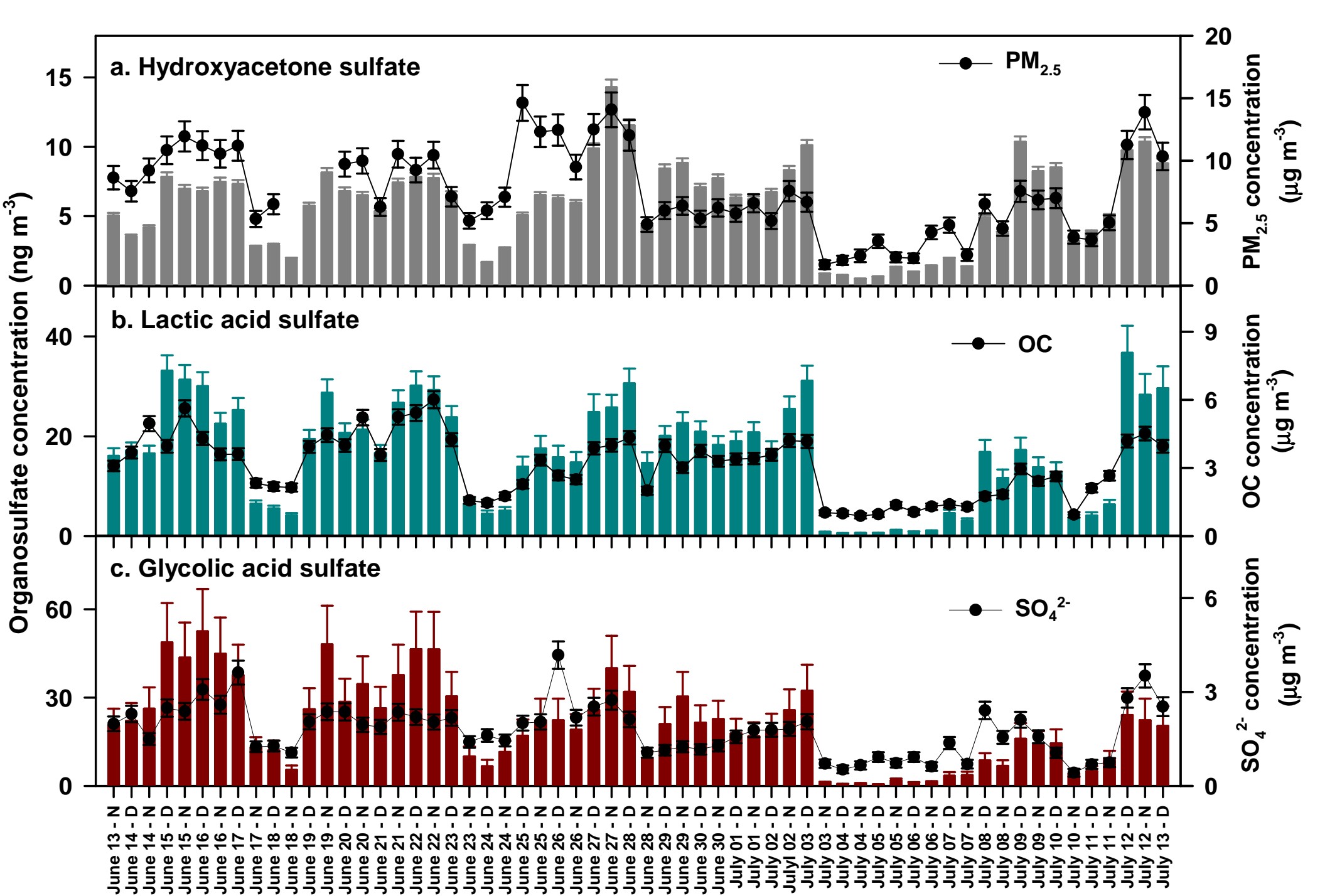

**Figure 2.**

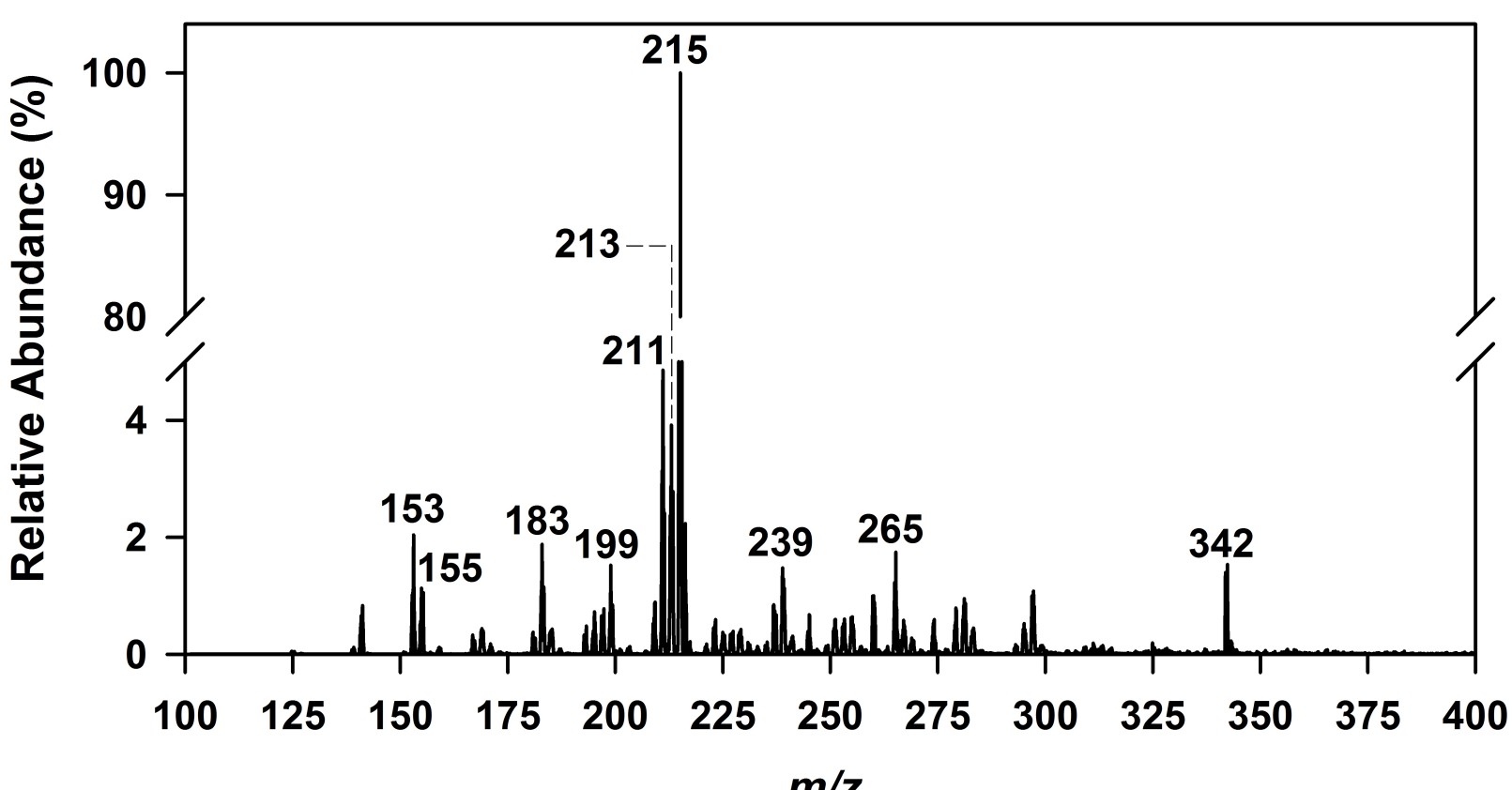

**Figure 3.**

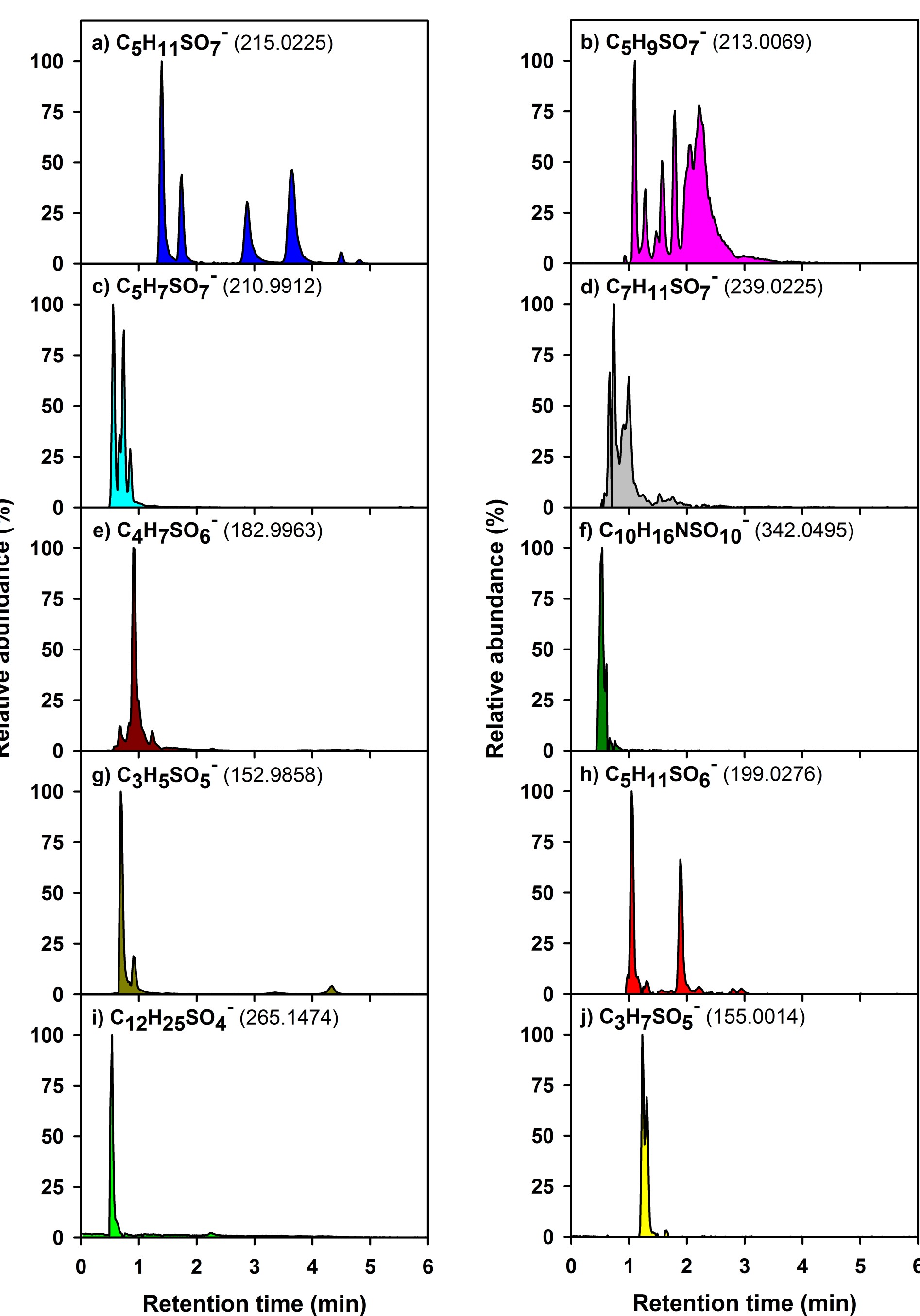