# Peer review of "Qualitative and Quantitative Analysis of Atmospheric Organosulfates in Centreville, Alabama"

_Atmospheric Chemistry and Physics, 2016_

## Referee Comment (RC1) · Anonymous Referee #1 · 11 Sep 2016

Summary and Recommendation:

This manuscript summarizes quantitative and semi-quantitative data obtained for organosulfates chemically characterized from PM2.5 samples collected from the main ground site (Centreville, AL) during the 2013 Southern Oxidant and Aerosol Study (SOAS). This study had 3 major goals: (1) to quantify select organosulfates that had authentic standards available using HILIC interfaced to ESI-triple quadrupole mass spectrometry; (2) assess for potential positive filter sampling artifacts of organosulfates; and (3) identify other major organosulfates that should be targets for future quantification once authentic standards are available. Analytically, this paper is very solid. The authors make a serious effort in understanding potential positive artifacts of organosul-

fates and find that they have fairly small artifacts. This is good to have these results in the literature.

This paper will certainly be of interest to the broader readership of ACP since organosulfates are good indicator compounds of mulitphase chemical reactions! However, there are some weaknesses that need to be improved upon before full publication in ACP. (Weakness 1) In some parts of the manuscript the writing is unclear or not explicit enough. I will point these out in my specific comments below. (Weakness 2) If your goal was to identify the major organosulfates at CTR during the 2013 SOAS study, I'm curious as to why only 4 days of sampling were considered? Why weren't the periods of intensive sampling included? From what I understand from this campaign (Budisulistiorini et al., 2015, ACP), chemical forecasts were made when biogenic VOCs and anthropogenic pollutants (sulfate) would be high. I believe the period chosen falls outside of these periods. Further, wouldn't analyzing most of the days for organosulfates also provide stronger statistics? (Weakness 3) In section 3.1 of the results and discussion, why wasn't more work done to investigate the potential sources (VOCs and/or their oxidation products as well as reactions) of these quantified organosulfates, especially since CTR had a wealth of gas and aerosol phase data? Since you focus on the quantification of these 4 organosulfates, it seems to me it would be interesting to at least examine potential correlations with other data sets to test previously proposed mechanisms for these products. That would add some more "beef" to the scientific discussion of these organosulfates. (Weakness 4) Have the authors considered adding into their discussion of the mass contribution of organosulfates quantified previously using authentic standards to the total OC/PM mass the data from Rattanavaraha et al. (2016, ACP, Table 5). That paper included the average MAE- and IEPOX-derived OSs quantified using the authentic standards for the CTR site. I think you can use these numbers to provide further insights into the potential overall mass contribution of these organosulfates (with yours here) to the total OC/PM2.5 mass. That seems like an important thing to do here. Once you add these in, how much closer do you get to the mass fractions of organosulfates reported by Tolocka and Turpin (2012, ES&T)?

(Weakness 5) For your qualitative discussion of other major organosulfates present at CTR, what about OSs that do not fragment to the m/z 97 ion in MS2? Prior work has shown that other important organosulfates, especially from monoterpenes (like m/z 294), may produce only the m/z 96 product ion (Surratt et al., 2008, JPCA) in MS2 spectra. I would at least acknowledge that you may be missing some important organosulfates since you focus your analyses only on those that produce the m/z 97 product ion in MS2 analyses.

Specific Comments:

1.) Abstract, Page 1, Lines 18-19: You should probably emphasize that this organosulfate is derived from mulitphase chemistry of IEPOX (Surratt et al., 2010, PNAS; Lin et al., 2012, ES&T).

2.) Introduction, Page 2, Lines 2-5: Should you be more specific and emphasize that PM2.5 has these adverse effects on human health and climate as well as contains most of the SOA?

3.) Introduction, Page 2, Line 4: I would insert "atmospheric oxidation" before "reactions"

4.) Introduction, Page 2, Lines 5-6: You should rephrase this sentence to be more correct. Maybe something like: "Organosulfates, which are produced from acid-catalyzed particle-phase reactions of gaseous oxidation products, such as epoxides (Lin et al., 2012, ES&T) and hydroperoxides (Mutzel et al., 2015, ES&T), contribute to SOA."

5.) Introduction, Page 2, Line 13: Now you switch to PM2.5. You should define this since this is its first use.

6.) Introduction, Page 2, Line 25: The beginning of this sentence should be reworded, possibly to "The most abundant organosulfates to be previously quantified include....."

7.) Introduction, Page 2, Line 32: change "instead" to "used"

8.) Introduction, Page 2, Line 32: Define the acronym "(-) ESI" for the first time here.

9.) Introduction, Page 3, Line 9: change ", however" to "; however, "

10.) Introduction, Page 3, Lines 18-19: Not sure how relevant this sentence is to the discussion here. I believe the Ehn et al. (2010, ACP) study could measure extremely low vapor pressure products in the gas phase (there still of course is an equilibrium between the gas and aerosol phase) such as the glycolic acid sulfate due to the high sensitivity of their CIMS instrument.

11.) Page 3, Line 27: Change "epoxides" to "epoxydiols"

12.) Section 2.2: In this section, I would be clear on which samples were analyzed. You should also be clear on why on these samples were extracted and analyzed for this study.

13.) Page 7, Line 12: Is this an average glycolic acid sulfate concentration from this BHM study or the upper limit? Please clarify.

---

## Referee Comment (RC2) · Anonymous Referee #2 · 3 Oct 2016

The manuscript by Hettiyadura et al. presents measured organosulphate (OS) concentrations in aerosol from the South East US from a four-day period during the SOAS campaign in the summer of 2013 at Centreville, Alabama. OS are an important contributor not necessarily due to their contribution to PM mass, but because they are the result of multi-phase processes and anthropogenic influence. The stated goals of the study are (i) quantification of OS (for which authentic standards are available) in PM2.5, (ii) assessment of filter sampling artefacts, and (iii) identifying major OS in Centreville.

The analytical work is very thorough using state-of-the-art methods and the finding on the filter artefacts will be important for future work on OS. Similarly, the progress toward identifying/ruling out isomers/functional groups is an important contribution. The main

concerns I have that should be addressed before publication is considered are clearer statements on the broader impact/significance beyond the analytical approach/work.

Major comments:

1. In order for the measurements to have significance beyond the very nice analytical method and artefact description and not simply be an anecdotal note of specific OS, it is critical to describe to what degree the very limited 4 day period was representative. As there was a plethora of measurements obtained at the Centreville site this should be easy. For example, where temperature, photochemical conditions, NOx conditions, amount of PM typical and, even more importantly, how variable were these conditions and is there any correlation with the observed OS variability shown in figure 1 (see point 2). Such a description of putting the measurements within the broader context of the SOAS campaign would help readers evaluate the broader significance of the observations described here.

2. It is stated that the work is complementary to that of Riva et al. 2016. However, it would be helpful if the similarities and differences with both the work/findings by Riva et al. 2016 and Rattanavaraha et al. 2016 would be stated more explicitly. For example, which OS were not observed by the two mentioned studies and how are the results similar and different? In fact, the time series in the Riva et al. 2016 (figure 4 of that manuscript) has a much longer dataset and it shows dramatic variability for organosulfate concentrations, which directly relates to point 1. In fact, a strong recommendation would be to collaborate with the Surratt group and use their much more extensive set of filter samples for the work described here.

3. It is stated that the work provides new insights for the major OS species in the SE US. Again, it would be helpful to explicitly state what the new insights are. For example, which of the major OS had not been identified before, and if they had been identified it would be useful to describe what additional new insight is gained for each of the major species. Clearly, such new insights exist, e.g., resulting from the analytical

approach such as ruling out carboxylic acid functional groups for some OS. Ending the manuscript by stating that there are new insights but not mentioning what they are could then be improved.

In summary, it would be helpful to make it easier for readers to identify clearly the novelty of the work/findings and the significance. To this end it may make sense to reorganize findings, e.g, (i) first show the 10 major OS, (ii) highlight the work to identify functional groups and isomers, which is a very nice and important contribution, and (iii) then discuss insights/recommendations. I think this may make it easier to recognize the significance, as the interesting new findings would not be interspersed within the "long" list of ten major OS. My second recommendation is to collaborate with the Surratt group using their extensive filter sample range, if possible.

Minor comments:

P.1 line 5: "from biogenic volatile…" As written it implies that only BVOCs form OS?

p.1 Line 13-4: "their VOC precursors" is a little vague, as isoprene is one of the VOC precursors for OS, but I don't think the authors are implying that isoprene reacts on the filters to form SO. It would be useful to clarify.

p.1 line 19: "Most of the ten…" Please be specific. How many?

p.2 line 2: "PM adversely affects … climate" This is a matter of debate. Some would say that PM positively affects climate due to counteracting greenhouse gas radiative effects. I would consider rephrasing

p.2 line 6-7: The authors could also consider the work of Liao et al. 2015 as it discusses acid effects. Currently, only ground based studies are cited.

p.2 line 12: I think it would be more specific to state that OS may be useful markers for one type of anthropogenic influence on SOA formation from biogenic VOCs, as there surely must be aspects of anthropogenic influence that the sulphate does not represent.

p.3 line 28: My understanding is that it is not clear whether the organosulfate is from methacrylic acid epoxide (MAE )or from hydroxymethyl-methyl-alpha-lactone (HMML), see Rattanavaraha et al. 2016?

p. 3 line 31-32: I think follow-up studies (Gallowy et al. 2011 and Liao et al. 2015) showed that glycolic acid sulphate was unlikely to result from (photochemical formation) from glyoxal and that the mechanism/source was unknown? Similarly, the formation of lactic acid sulphate from methylglyoxal seems mechanistically challenging.

p. 4 line 25: Please state the total organic carbon content as resistivity does not address the content of uncharged organic compounds.

p. 5 line 7: front QFF. Although it is fairly clear, defining better what the front QFF is would be useful (actual sample QFF?)

p. 6 line 29-30: If the mass range was 400 Da, why consider up to 500 carbon atoms, corresponding to 6000 Da?

p.7 line 18-20. Please put these results in context with the ones previously mentioned by Tolocka and Turpin 2012).

p.8 line 24-27. I don't understand how the second sentence follows from the first: (i) there is some OS formation on the acidified filters, (ii) SOA is acidic enough and has high enough sulphate that these are not limiting factors. Are the authors implying that the gas-phase is already depleted of precursors or what is then limiting?

p. 9 line 3: Does "negative sampling artefact" imply destruction of the OS in question? It would be helpful to clarify and explain

p.9 line 23-24. Does this mean that not all condensed-phase is evaporated when using water with little acetonitril, i.e., that liquid water remains or just a few H2O molecules on the OS and are such signals seen? Could it be that the water takes some of the charge and that or in some other way suppresses/reduces the ionization efficiency of the OS? Please explain this effect better.

[Figure]

p.11 line 20. Given the very high vapour pressure of MVK and MACR is it reasonable to assume that they are present in any significant concentration in PM in the first place to be able to oligomerize?

Technical comments:

There are some grammatical errors, e.g., noun-verb agreement, and the manuscript could benefit from some proof reading. Some examples: p.2 line 13: "SAO accounts for a significant . . . and suggested. . ." p.2 Line 25 : "Among them are most abundant organosulfate has been . . ." p.2 Line 31-32: "In the absence of authentic standards, surrogate standards are commonly instead, but can" p.3 line 9: "have been discussed", "are discussed" is perhaps more suitable p.8 line 20 "14-15: "the potential . . . were assessed" p.11 line "forms" instead of "form" p. 14 line 9. "of" instead of "to"?

There are some places where it is not quite clear what is meant, e.g.:

p.2 line 16-17: Stating that high sulphate etc. make the atmosphere subject to anthropogenic influence sounds a little odd to me. Do they actually not directly represent the anthropogenic influence?

p.8 line 9-11: "The very minor influence of . . . may be promoted . . ." I am not sure what promoting a minor influence means, and the "and possibly temperature" also seems a little out of place.

p. 3 line 8: MS2 has not been defined, I think. Some explanation of this method would be useful for readers to understand the following statements.

Galloway, M. M., Loza, C. L., Chhabra, P. S., Chan, A. W. H., Yee, L. D., Seinfeld, J. H., Keutsch, F. N.: Analysis of photochemical and dark glyoxal uptake: Implications for SOA formation, Geophys. Res. Lett. 38, L17811, doi:10.1029/2011GL048514, 2011.

Liao, J., Froyd, J. K. D., Murphy, D. M., Keutsch, F. N., Yu, G., Wennberg, P. O., St. Clair, J. Crounse, J. D., Wisthaler, A., Mikoviny, T., Jimenez, J.-L., Campuzano Jost, P., Day, D. A., Hu, W, Ryerson, T. B., Pollack, I. B., Peischl, J., Anderson, B. E., Ziemba, L. D.,

Blake, D. R., Meinardi, S., Diskin, G.: Airborne measurements of organosulfates over the continental US, J. Geophys. Res. 120, 2990-3005, doi:10.1002/2014JD022378, 2015.

Rattanavaraha, W., Chu, K., Budisulistiorini, S. H., Riva, M., Lin, Y. H., Edgerton, E. S., Baumann, K., Shaw, S. L., Guo, H., King, L., Weber, R. J., Neff, M. E., Stone, E. A., Offenberg, J. H., Zhang, Z., Gold, A., and Surratt, J. D.: Assessing the impact of anthropogenic pollution on isoprene-derived secondary organic aerosol formation in PM2.5 collected from the Birmingham, Alabama, ground site during the 2013 Southern Oxidant and Aerosol Study, Atmos. Chem. Phys., 16, 4897-4914, doi:10.5194/acp-16-4897-2016, 2016.

Riva, M., Budisulistiorini, S. H., Zhang, Z., Gold, A., and Surratt, J. D.: Chemical characterization of secondary organic aerosol constituents from isoprene ozonolysis in the presence of acidic aerosol, Atmos. Environ., 130, 5-13, doi:http://dx.doi.org/10.1016/j.atmosenv.2015.06.027, 2016.

---

## Author Comment (AC1) · 21 Nov 2016

Anonymous Referee # 1, Summary and Recommendation: "This manuscript summarizes quantitative and semi-quantitative data obtained for organosulfates chemically characterized from PM2.5 samples collected from the main ground site (Centreville, AL) during the 2013 Southern Oxidant and Aerosol Study (SOAS). This study had 3 major goals: (1) to quantify select organosulfates that had authentic standards available using HILIC interfaced to ESI-triple quadrupole mass spectrometry; (2) assess for potential positive filter sampling artifacts of organosulfates; and (3) identify other major organosulfates that should be targets for future quantification once authentic standards are available. Analytically, this paper is very solid. The authors make a serious effort

in understanding potential positive artifacts of organosulfates and find that they have fairly small artifacts. This is good to have these results in the literature. This paper will certainly be of interest to the broader readership of ACP since organosulfates are good indicator compounds of mulitphase chemical reactions! However, there are some weaknesses that need to be improved upon before full publication in ACP."

Response to Anonymous Referee # 1 Summary and Recommendations: We thank the referee for their thoughtful and valuable insights. We agree with their summary of the scope of this work. We have revised this paper addressing each of the weaknesses and specific comments, point by point as indicated below.

Anonymous Referee # 1 Weakness 1: "In some parts of the manuscript the writing is unclear or not explicit enough. I will point these out in my specific comments below."

Response to Anonymous referee # 1 Weakness 1: We have provided responses and revisions to the referee's suggestions on writing in specific comments 1 – 13.

Anonymous Referee # 1 Weakness 2: "If your goal was to identify the major organosulfates at CTR during the 2013 SOAS study, I'm curious as to why only 4 days of sampling were considered? Why weren't the periods of intensive sampling included? From what I understand from this campaign (Budisulistiorini et al., 2015, ACP), chemical forecasts were made when biogenic VOCs and anthropogenic pollutants (sulfate) would be high. I believe the period chosen falls outside of these periods. Further, wouldn't analyzing most of the days for organosulfates also provide stronger statistics?"

Response to Referee # 1 Weakness 2: This comment has brought about two major changes to the manuscript. First, we provide a more detailed description of the subset of samples studied for sampling artifacts and how these days relate to average conditions during SOAS. We also note that the subset of samples (07 – 11 July, 2013) overlaps with the 4th intensive sampling period during SOAS (9-14 July). Second, we have expanded the range of quantitative data presented to include 13 June – 13 July as described in response to the next comment.

Section 2.2 has been revised to read: "The positive filter sampling artifacts associated with the three most abundant organosulfates quantified in Sect. 3.1 (glycolic acid sulfate, lactic acid sulfate and hydroxyacetone sulfate, respectively) were assessed from 07 – 11 July, 2013. This time period followed several days with rain, thus had slightly lower average PM2.5 (5.24 ± 1.68 $\mu$g m-3), OC (2.00 ± 0.67 $\mu$g m-3), sulfate (1.26 ± 0.66 $\mu$g m-3) and organosulfate concentrations relative to the average PM2.5 (7.52 ± 3.41 $\mu$g m-3), OC (3.07 ± 1.35 $\mu$g m-3), sulfate (1.78 ± 0.81 $\mu$g m-3) and organosulfate concentrations measured during SOAS in Centreville (Fig. 1 and Table 1). Within the studied subset of days, the 09 July daytime and nighttime, and 10 July daytime concentrations (Fig. 1) were similar to the average conditions observed during SOAS, and are considered to be most representative of the average conditions at Centreville during SOAS."

This text has been added to section 3.3, page 9 at line 8: "This analysis was applied to samples collected from 07 – 11 July, 2013, with a focus on the 10 July daytime sample with levels of PM2.5 (7.01 ± 0.80 $\mu$g m-3), OC (2.63 ± 0.21 $\mu$g m-3), sulfate (1.06 ± 0.17 $\mu$g m-3) and organosulfates (Fig. 1) near to the study average (Sect. 3.2 and Table 1)."

Anonymous Referee # 1 Weakness 3: "In section 3.1 of the results and discussion, why wasn't more work done to investigate the potential sources (VOCs and/or their oxidation products as well as reactions) of these quantified organosulfates, especially since CTR had a wealth of gas and aerosol phase data? Since you focus on the quantification of these 4 organosulfates, it seems to me it would be interesting to at least examine potential correlations with other data sets to test previously proposed mechanisms for these products. That would add some more "beef" to the scientific discussion of these organosulfates."

Response to Referee # 1 Weakness 3: As suggested, we have extended the data presented from 7 – 11, July 2013 to 13 June – 13 July, 2013; with this larger dataset, we provide a more in-depth correlation analysis with VOC precursors and other PM constituents measured in Centreville, during SOAS 2013. Accordingly, we have revised our objectives to include correlations and a paragraph was added to section at 3.2 discussing the correlation results. Also, note that by adding more measurements required minor updates to numerical values in Tables 1 and 3 (where the latter was previously Table 2).

The text that has been added to page 7, section 3.2, line 23: "Correlations of hydroxyacetone sulfate, lactic acid sulfate and glycolic acid sulfate with co-located gas and aerosol measurements were used to gain insights to their potential precursors and conditions conducive to their formation (Table 2). Strong inter-correlations were observed for these organosulfates suggesting that they have common precursors and/or formation pathways. All three species had higher correlations with formaldehyde, MACR and glyoxal relative to isoprene, ISOPOOH and IEPOX that are low NOx oxidation products of isoprene (Bates et al., 2016; Krechmer et al., 2015)), as well as MVK and isoprene nitrates (ISOPN) that are high NOx oxidation product (Xiong et al., 2015)). While MVK, MACR, glyoxal and formaldehyde may be either biogenic or anthropogenic in origin, they primarily form from isoprene oxidation in SE US during summer (Xiong et al., 2015; Kaiser et al., 2015). Previous studies have shown that MVK, MACR, glyoxal and formaldehyde form in higher yields when isoprene was oxidized under high NOx (Kaiser et al., 2015; Liu et al., 2013). Of MVK and MACR, MACR is the major SOA precursor form from isoprene oxidation under high NOx conditions (Surratt et al., 2006; Kroll et al., 2006; Surratt et al., 2010). Thus the higher correlations with formaldehyde, MACR and glyoxal relative to other VOC precursors suggest that these organosulfates are enhanced by high NOx conditions.

All three species had moderate to strong correlations with sulfate, but not with liquid water content or acidity, suggesting that neither aerosol water nor aerosol acidity limit organosulfate formation. Similar correlations were reported at Centreville for isoprene derived SOA, and were attributed to variation of sulfate compared to consistently high aerosol acidity and high relative humidity observed during SOAS 2013 (Xu et al.,

2015). Further, these correlations are consistent across other SOAS ground sites (Rattanavaraha et al., 2016; Budisulistiorini et al., 2015) indicating that the association of organosulfates with sulfate is a regional characteristic. The correlations of organosulfates derived from isoprene and sulfate in the SE US, suggests that sulfate is a key factor that influences biogenic SOA formation."

Anonymous Referee # 1 Weakness 4, Page 7, Section 3.1: "Have the authors considered adding into their discussion of the mass contribution of organosulfates quantified previously using authentic standards to the total OC/PM mass the data from Rattanavaraha et al. (2016, ACP, Table 5). That paper included the average MAE- and IEPOX-derived OSs quantified using the authentic standards for the CTR site. I think you can use these numbers to provide further insights into the potential overall mass contribution of these organosulfates (with yours here) to the total OC/PM2.5 mass. That seems like an important thing to do here. Once you add these in, how much closer do you get to the mass fractions of organosulfates reported by Tolocka and Turpin (2012, ES&T)?"

Response to Referee # 1 Weakness 4: As suggested by the reviewer, we have expanded our discussion to include the total mass contribution of organosulfates quantified in Centreville using authentic standards and the mass closure achieved when combining our results with those of Rattanavaraha et al. (2016).

Page 7, section 3.1, lines 18 – 22 originally read: "The total contribution of the organosulfates quantified using authentic standards accounted for less than 0.5 % of PM2.5 and less than 0.3 % of OC (Table 1). Meanwhile, organosulfates are estimated to contribute 1-2 % of PM2.5 and 5-10 % OC in Eastern US (Shakya and Peltier, 2015). Therefore, the organosulfates quantified against authentic standards account for a minority of the total organosulfates, while other organosulfates likely comprise the majority of this class of compounds in Centreville, AL (as discussed in Sect. 3.3)."

This text has been revised to read: "The total contribution of the organosulfates quantified using authentic standards was less than 0.3 % of OC (Table 1). Meanwhile, the estimated upper bound contribution of organosulfates to organic matter (OM) is 5.0 – 9.3 % in the SE US (Tolocka and Turpin, 2012). Assuming OM/OC of 1.8 (Tolocka and Turpin, 2012), the calculated contribution of the organosulfates quantified in this study comprise 0.7 % of OM. Measurements of 2-methyltetrol sulfates reported by Ratanva-hara et al. (2016) for Centreville had a mean concentration of 207.1 ng m-3 and were estimated to account for 3.7% while 2-methylglyceric acid sulfate had a mean concentration of 10.2 ng m-3 and accounted for 0.2% of OM, y considering the average OC concentration of 3.07 ug m-3 and an OM/OC ratio of 1.8. Together, the organosulfates quantified against authentic standards in Centreville accounts for 4.7 % of OM. Additional species that contribute significantly to MS2 organosulfate signals are qualitatively and semi-quantitatively examined in Sect. 3.3.

Anonymous Referee # 1 Weakness 5: "For your qualitative discussion of other major organosulfates present at CTR, what about OSs that do not fragment to the m/z 97 ion in MS2? Prior work has shown that other important organosulfates, especially from monoterpenes (like m/z 294), may produce only the m/z 96 product ion (Surratt et al., 2008, JPCA) in MS2 spectra. I would at least acknowledge that you may be missing some important organosulfates since you focus your analyses only on those that produce the m/z 97 product ion in MS2 analyses."

Response to Referee # 1 Weakness 5: We thank the referee for pointing this out. We have analyzed the organosulfates that fragmented to m/z 96, but initially did not include these results because of the low signal (2 % of the precursors of m/z 97). However, this comment suggests that the community would be interested in our findings from our studies of precursors to m/z 96 and thus we have added them to the revised manuscript.

Results from MS2 scans of precursors are shown in Figure S4 and Table S2. In addition, these results have been discussed in added to section 3.3. In particular, the following text has been added: "In addition, a nitro-oxy organosulfate, C5H10NSO9-

(260.0076; Fig. S4) contributed up to 5.4 % of the m/z 96 precursor ion signal (Table S1) and is also associated with isoprene (Surratt et al., 2008; Gómez-González et al., 2008)."

"Monoterpene-derived nitro-oxy organosulfates were particularly responsive to precursors of m/z 96; C10H16NSO10- Âň (342.0495), C10H16NSO8- (310.0597 (Ma et al., 2014; Surratt et al., 2008)) and C10H16NSO7- (294.0647) (Table S1 and Fig. S4). The nitro-oxy organosulfate C10H16NSO7- (294.0647) accounted for 25 % of the total m/z 96 signal in PM2.5 sample collected during nighttime on 10 July 2013 (Table S1). This semi-quantitative result is consistent with prior field studies reported m/z 294 as the most abundant nitro-oxy organosulfate in SE US, particularly during night time (Gao et al., 2006; Surratt et al., 2008).

Other major organosulfate signals identified from m/z 96, were C4H7SO4- (151.0065), C3H5SO4- (136.9909) and C5H8NO8S- (241.9971), were not previously reported in the atmosphere (Table S2). Based on the molecular formula and double bond equivalence (Table S1), m/z 151 is suggested as a methylallyl sulfate, m/z 137 may be allyl sulfate and m/z 242 may be a nitro-oxy organosulfate with a carbonyl group. However, the precursors to these organosulfates are unknown."

Anonymous Referee # 1 Specific Comments:

Referee # 1 Specific Comment 1) Abstract, Page 1, Lines 18-19: "You should probably emphasize that this organosulfate is derived from multiphase chemistry of IEPOX (Surratt et al., 2010, PNAS; Lin et al., 2012, ES&T)."

Response to Referee # 1 Specific Comment 1: While we agree with the reviewer, we do not think the abstract is the appropriate place to convey results from prior studies. Instead, this information has been integrated into the introduction and discussion of 2-methyltetrol sulfate results.

Referee # 1 Specific Comment 2) Introduction, Page 2, Lines 2-5: "Should you be

more specific and emphasize that PM2.5 has these adverse effects on human health and climate as well as contains most of the SOA?"

Response to Referee # 1 Specific Comment 2: We agree with the reviewer and have revised the text accordingly.

The Introduction, Page 2, Lines 2-5 originally read: "Atmospheric particulate matter (PM) adversely affects human health and climate (Anderson et al., 2011; Kim et al., 2015; Rosenfeld et al., 2014; Levy et al., 2013). A significant fraction of PM is comprised of secondary organic aerosols (SOA) (Zhang et al., 2011) that form from reactions of volatile organic compounds (VOC) yielding semi-volatile products that partition to the aerosol phase."

This text has been revised to read: "Atmospheric fine particulate matter (PM2.5; particles $\leq 2.5$ $\mu$m in aerodynamic diameter) adversely affects human health (Valavanidis et al., 2008; Anderson et al., 2011; Kim et al., 2015) and influences the Earth's climate via direct and indirect radiative forcing (Novakov and Penner, 1993; Haywood and Boucher, 2000). A significant fraction of PM2.5 organic matter is secondary in origin (Zhang et al., 2011), and forms by atmospheric oxidation reactions of volatile organic compounds (VOC) and partitioning of reaction products to the aerosol phase (Hallquist et al., 2009)."

Referee # 1 Specific Comment 3) Introduction, Page 2, Line 4: "I would insert "atmospheric oxidation" before "reactions""

Response to Referee # 1 Specific Comment 3: Introduction, Page 2, Lines 4: We have revised this sentence as indicated in the response to specific comment 2 (last sentence).

Referee # 1 Specific Comment 4) Introduction, Page 2, Lines 5-6: "You should rephrase this sentence to be more correct. Maybe something like: "Organosulfates, which are produced from acid-catalyzed particle-phase reactions of gaseous oxidation products,

such as epoxides (Lin et al., 2012, ES&T) and hydroperoxides (Mutzel et al., 2015, ES&T), contribute to SOA."

Response to Referee # 1 Specific Comment 4: The introduction, Page 2, Lines 5-6 originally read: "Among SOA products are organosulfates, which are produced in the presence of sulfate aerosol and are particularly enhanced under acidic conditions (Surratt et al., 2007b; Surratt et al., 2010; Surratt et al., 2008; Surratt et al., 2007a)."

The text has been revised to read: "Among secondary organic aerosols (SOA) are organosulfates, which are mainly produced from acid-catalyzed particle-phase reactions of gaseous oxidation products such as epoxides (Lin et al., 2012) and hydroperoxides (Mutzel et al., 2015) with sulfate (Surratt et al., 2007b; Surratt et al., 2010; Surratt et al., 2008; Surratt et al., 2007a; Liao et al., 2015)."

Referee # 1 Specific Comment 5) Introduction, Page 2, Line 13: "Now you switch to PM2.5. You should define this since this is its first use."

Response to referee # 1 Specific Comment 5: We have implemented this suggestion and the revised text is provided in response to Referee # 1 Specific Comment 2.

Referee # 1 Specific Comment 6) Introduction, Page 2, Line 25: "The beginning of this sentence should be reworded, possibly to "The most abundant organosulfates to be previously quantified include....."""

Response to referee # 1 Specific Comment 6: We have revised this sentence as suggested.

Referee # 1 Specific Comment 7) Introduction, Page 2, Line 32: "change "instead" to "used"".

Response to referee # 1 Specific Comment 7: We have revised this sentence as suggested.

Referee # 1 Specific Comment 8) Introduction, Page 2, Line 32: "Define the acronym

"(-) ESI" for the first time here."

Response to Referee # 1 Specific Comment 8: Introduction, Page 2, Line 32: We have defined '(-) ESI', as indicated in the response to specific comment 7.

Referee # 1 Specific Comment 9) Introduction, Page 3, Line 9: "change ", however" to "; however, ""

Response to Referee # 1 Specific Comment 9: As suggested, we have changed the comma to a semicolon.

Referee # 1 Specific Comment 10) Introduction, Page 3, Lines 18-19: "Not sure how relevant this sentence is to the discussion here. I believe the Ehn et al. (2010, ACP) study could measure extremely low vapor pressure products in the gas phase (there still of course is an equilibrium between the gas and aerosol phase) such as the glycolic acid sulfate due to the high sensitivity of their CIMS instrument."

Response to Referee # 1 Specific Comment 10: We agree with the referee that detection of gaseous glycolic acid sulfate in Ehn et al., 2010 emphasize the high sensitivity of their detection method (APi-ToF) to extremely low concentrations of glycolic acid sulfate in the gas phase that is in equilibrium with aerosol phase. Consequently, we have removed this sentence from the text.

Referee # 1 Specific Comment 11) Introduction, Page 3, Line 27: "Change "epoxides" to "epoxydiols""

Response to Referee # 1 Specific Comment 11: We have revised this sentence as suggested.

Referee # 1 Specific Comment 12) Section 2.2: "In this section, I would be clear on which samples were analyzed. You should also be clear on why on these samples were extracted and analyzed for this study."

Response to referee # 1 specific comment 12: As suggested by the reviewer, we have

indicated which samples were analyzed for quantification of organosulfates and for the sampling artifacts study in section 2.2. The reason for why these samples were analyzed to identify major organosulfates in Centreville is given in response to weakness 2.

Referee # 1 Specific Comment 13) Page 7, Line 12: "Is this an average glycolic acid sulfate concentration from this BHM study or the upper limit? Please clarify."

Response to referee #1 specific comment 13: Section 3.1, Page 7, Line 12 originally read: "At the nearby Birmingham, AL which is an industrial and residential site even higher glycolic acid sulfate concentrations (75.2 ng m-3) were reported from 01 June – 15 July, 2013 during SOAS (Rattanavaraha et al., 2016) with a mean concentration of 26.2 ng m-3."

This text is revised to read: "At the nearby Birmingham, AL site during SOAS, similar organosulfate concentrations were reported: glycolic acid sulfate averaged 26.2 ng m-3 and had a maximum value of 75.2 ng m-3. . ."

References Anderson, J. O., Thundiyil, J. G., and Stolbach, A.: Clearing the Air: A Review of the Effects of Particulate Matter Air Pollution on Human Health, J. Med. Toxicol., 8, 166-175, doi:10.1007/s13181-011-0203-1, 2011. Bates, K. H., Nguyen, T. B., Teng, A. P., Crounse, J. D., Kjaergaard, H. G., Stoltz, B. M., Seinfeld, J. H., and Wennberg, P. O.: Production and Fate of C4 Dihydroxycarbonyl Compounds from Isoprene Oxidation, The Journal of Physical Chemistry A, 120, 106-117, doi:10.1021/acs.jpca.5b10335, 2016.

[revised manuscript text omitted]

**Table 2**: Correlations of hydroxyacetone sulfate, lactic acid sulfate and glycolic acid sulfate with $PM_{2.5}$, isoprene, high NOx isoprene oxidation products such as isoprene hydroxyl nitrates (ISOPN), methacrolein (MACR), methylvinyl ketone (MVK), glyoxal, formaldehyde, hydroxyacetone and glycolaldehyde, low NOx isoprene oxidation products such as isoprene hydroxyl hydroperoxide (ISOPOOH) and isoprene dihydroxy epoxides (IEPOX) and PM constituents such as sulfate, aerosol water and aerosol acidity in Centreville, AL during SOAS 2013. Underlined correlation coefficients are statistically significant at 95 % confidence interval ($p \leq 0.05$).

| VOC precursor/PM constituent | Number of samples | Pearson correlation coefficient (r) | | |
|---|---|---|---|---|
| | | Glycolic acid sulfate | Lactic acid sulfate | Hydroxyacetone sulfate |
| Lactic acid sulfate | 60 | | | 0.86 |
| Glycolic acid sulfate | 60 | | 0.88 | 0.71 |
| Formaldehyde | 60 | 0.73 | 0.76 | 0.69 |
| Sulfate | 60 | 0.69 | 0.74 | 0.63 |
| Hydroxyacetone | 42 | 0.68 | 0.70 | 0.63 |
| MACR | 59 | 0.67 | 0.67 | 0.59 |
| Glyoxal | 60 | 0.59 | 0.64 | 0.56 |
| ISOPOOH | 38 | 0.52 | 0.48 | 0.32 |
| Glycolaldehyde | 39 | 0.45 | 0.48 | 0.36 |
| Isoprene | 59 | 0.44 | 0.40 | 0.45 |
| MVK | 59 | 0.30 | 0.43 | 0.35 |
| ISOPN | 42 | 0.32 | 0.40 | 0.30 |
| IEPOX | 38 | 0.40 | 0.41 | 0.14 |
| Aerosol water | 56 | 0.32 | 0.26 | 0.33 |
| Aerosol acidity | 49 | -0.14 | 0.13 | 0.20 |

**Fig. 1.**

[Figure]

Fig. 2.

[Figure]

**Fig. 3.**

**Table S2**: Other organosulfates that were detected among the ten greatest intensity signals in precursors of $m/z$ 97 ($HSO_4^-$) for $PM_{2.5}$ samples collected from 07-11 July 2013 during SOAS by HILIC-TQD and their HR-MS characterization using HILIC-ToF.

| Molecular formula [M-H]⁻ | Double bond equivalence(s) | Monoisotopic mass | Error in observed mass (mDa) | $t_R$ HILIC-TOF (min) |
|---|---|---|---|---|
| $C_5H_7SO_6^-$ [a] | 2.5 | 194.9963 | 0.5 | 0.57 |
| | | | 1.5 | 0.74 |
| $C_7H_{11}SO_6^-$ [b] | 2.5 | 223.0276 | -0.9 | 0.51 |
| | | | -0.8 | 0.65 |
| | | | 1.0 | 0.80 |
| | | | -0.6 | 1.02 |
| | | | -0.8 | 1.16 |
| $C_7H_9SO_7^-$ [a] | 3.5 | 237.0069 | 1.1 | 0.65 |
| $C_{10}H_{15}SO_7^-$ [b] | 3.5 | 279.0538 | -3.7 | 0.54 |
| | | | 0.8 | 0.80 |
| $C_{10}H_{17}SO_7^-$ [b] | 2.5 | 281.0695 | -4.3 | 0.59 |
| | | | -8.8 | 0.80 |
| $C_{10}H_{17}SO_8^-$ [b] | 2.5 | 297.0644 | -1.7 | 0.74 |
| | | | -3.6 | 1.08 |
| | | | 0.1 | 1.85 |

[a] VOC precursors are unknown, although these $m/z$ have been previously identified in rain water (Altieri et al., 2009) and cloud water (Boone et al., 2015)
[b] Monoterpenes have been identified as VOC precursors to these $m/z$ (Surratt et al., 2008)

**Fig. 4.**

---

## Author Response (AR1)

**Anonymous Referee # 1, Summary and Recommendation:** "*This manuscript summarizes quantitative and semi-quantitative data obtained for organosulfates chemically characterized from PM$_{2.5}$ samples collected from the main ground site (Centreville, AL) during the 2013 Southern Oxidant and Aerosol Study (SOAS). This study had 3 major goals: (1) to quantify select organosulfates that had authentic standards available using HILIC interfaced to ESI-triple quadrupole mass spectrometry; (2) assess for potential positive filter sampling artifacts of organosulfates; and (3) identify other major organosulfates that should be targets for future quantification once authentic standards are available. Analytically, this paper is very solid. The authors make a serious effort in understanding potential positive artifacts of organosulfates and find that they have fairly small artifacts. This is good to have these results in the literature.*
*This paper will certainly be of interest to the broader readership of ACP since organosulfates are good indicator compounds of multiphase chemical reactions! However, there are some weaknesses that need to be improved upon before full publication in ACP.*"

Response to Anonymous Referee # 1 Summary and Recommendations: We thank the referee for their thoughtful and valuable insights. We agree with their summary of the scope of this work. We have revised this paper addressing each of the weaknesses and specific comments, point by point as indicated below.

**Anonymous Referee # 1 Weakness 1:** "*In some parts of the manuscript the writing is unclear or not explicit enough. I will point these out in my specific comments below.*"

Response to Anonymous referee # 1 Weakness 1: We have provided responses and revisions to the referee's suggestions on writing in specific comments 1 – 13.

**Anonymous Referee # 1 Weakness 2**: "*If your goal was to identify the major organosulfates at CTR during the 2013 SOAS study, I'm curious as to why only 4 days of sampling were considered? Why weren't the periods of intensive sampling included? From what I understand from this campaign (Budisulistiorini et al., 2015, ACP), chemical forecasts were made when biogenic VOCs and anthropogenic pollutants (sulfate) would be high. I believe the period chosen falls outside of these periods. Further, wouldn't analyzing most of the days for organosulfates also provide stronger statistics?*"

Response to Referee # 1 Weakness 2: This comment has brought about two major changes to the manuscript. First, we provide a more detailed description of the subset of samples studied for sampling artifacts and how these days relate to average conditions

during SOAS. We also note that the subset of samples (07 – 11 July, 2013) overlaps with the 4[th] intensive sampling period during SOAS (9-14 July). Second, we have expanded the range of quantitative data presented to include 13 June – 13 July as described in response to the next comment.

Section 2.2 has been revised to read: "The positive filter sampling artifacts associated with the three most abundant organosulfates quantified in Sect. 3.1 (glycolic acid sulfate, lactic acid sulfate and hydroxyacetone sulfate, respectively) were assessed from 07 – 11 July, 2013. This time period followed several days with rain, thus had slightly lower average $PM_{2.5}$ (5.24 ± 1.68 µg m$^{-3}$), OC (2.00 ± 0.67 µg m$^{-3}$), sulfate (1.26 ± 0.66 µg m$^{-3}$) and organosulfate concentrations relative to the average $PM_{2.5}$ (7.52 ± 3.41 µg m$^{-3}$), OC (3.07 ± 1.35 µg m$^{-3}$), sulfate (1.78 ± 0.81 µg m$^{-3}$) and organosulfate concentrations measured during SOAS in Centreville (Fig. 1 and Table 1). Within the studied subset of days, the 09 July daytime and nighttime, and 10 July daytime concentrations (Fig. 1) were similar to the average conditions observed during SOAS, and are considered to be most representative of the average conditions at Centreville during SOAS."

This text has been added to section 3.3, page 9 at line 8: "This analysis was applied to samples collected from 07 – 11 July, 2013, with a focus on the 10 July daytime sample with levels of $PM_{2.5}$ (7.01 ± 0.80 µg m$^{-3}$), OC (2.63 ± 0.21 µg m$^{-3}$), sulfate (1.06 ± 0.17 µg m$^{-3}$) and organosulfates (Fig. 1) near to the study average (Sect. 3.2 and Table 1)."

**Anonymous Referee # 1 Weakness 3**: "*In section 3.1 of the results and discussion, why wasn't more work done to investigate the potential sources (VOCs and/or their oxidation products as well as reactions) of these quantified organosulfates, especially since CTR had a wealth of gas and aerosol phase data? Since you focus on the quantification of these 4 organosulfates, it seems to me it would be interesting to at least examine potential correlations with other data sets to test previously proposed mechanisms for these products. That would add some more "beef" to the scientific discussion of these organosulfates.*"

Response to Referee # 1 Weakness 3: As suggested, we have extended the data presented from 7 – 11, July 2013 to 13 June – 13 July, 2013; with this larger dataset, we provide a more in-depth correlation analysis with VOC precursors and other PM constituents measured in Centreville, during SOAS 2013. Accordingly, we have revised our objectives to include correlations and a paragraph was added to section at 3.2 discussing the correlation results. Also, note that by adding more measurements required minor updates to numerical values in Tables 1 and 3 (where the latter was previously Table 2).

The text that has been added to page 7, section 3.2, line 23:

"Correlations of hydroxyacetone sulfate, lactic acid sulfate and glycolic acid sulfate with co-located gas and aerosol measurements were used to gain insights to their potential precursors and conditions conducive to their formation (Table 2). Strong inter-correlations were observed for these organosulfates suggesting that they have common precursors and/or formation pathways. All three species had higher correlations with formaldehyde, MACR and glyoxal relative to isoprene, ISOPOOH and IEPOX that are low $NO_x$ oxidation products of isoprene (Bates et al., 2016; Krechmer et al., 2015)), as well as MVK and isoprene nitrates (ISOPN) that are high $NO_x$ oxidation product (Xiong et al., 2015)). While MVK, MACR, glyoxal and formaldehyde may be either biogenic or anthropogenic in origin, they primarily form from isoprene oxidation in SE US during summer (Xiong et al., 2015; Kaiser et al., 2015). Previous studies have shown that MVK, MACR, glyoxal and formaldehyde form in higher yields when isoprene was oxidized under high $NO_x$ (Kaiser et al., 2015; Liu et al., 2013). Of MVK and MACR, MACR is the major SOA precursor form from isoprene oxidation under high NOx conditions (Surratt et al., 2006; Kroll et al.,

2006; Surratt et al., 2010). Thus the higher correlations with formaldehyde, MACR and glyoxal relative to other VOC precursors suggest that these organosulfates are enhanced by high $NO_x$ conditions.

All three species had moderate to strong correlations with sulfate, but not with liquid water content or acidity, suggesting that neither aerosol water nor aerosol acidity limit organosulfate formation. Similar correlations were reported at Centreville for isoprene derived SOA, and were attributed to variation of sulfate compared to consistently high aerosol acidity and high relative humidity observed during SOAS 2013 (Xu et al., 2015). Further, these correlations are consistent across other SOAS ground sites (Rattanavaraha et al., 2016; Budisulistiorini et al., 2015) indicating that the association of organosulfates with sulfate is a regional characteristic. The correlations of organosulfates derived from isoprene and sulfate in the SE US, suggests that sulfate is a key factor that influences biogenic SOA formation."

**Anonymous Referee # 1 Weakness 4,** Page 7, Section 3.1: "*Have the authors considered adding into their discussion of the mass contribution of organosulfates quantified previously using authentic standards to the total OC/PM mass the data from Rattanavaraha et al. (2016, ACP, Table 5). That paper included the average MAE- and IEPOX-derived OSs quantified using the authentic standards for the CTR site. I think you can use these numbers to provide further insights into the potential overall mass contribution of these organosulfates (with yours here) to the total OC/PM$_{2.5}$ mass. That seems like an important thing to do here. Once you add these in, how much closer do you get to the mass fractions of organosulfates reported by Tolocka and Turpin (2012, ES&T)?*"

Response to Referee # 1 Weakness 4: As suggested by the reviewer, we have expanded our discussion to include the total mass contribution of organosulfates quantified in Centreville using authentic standards and the mass closure achieved when combining our results with those of Rattanavaraha et al. (2016).

Page 7, section 3.1, lines 18 – 22 originally read: "The total contribution of the organosulfates quantified using authentic standards accounted for less than 0.5 % of PM$_{2.5}$ and less than 0.3 % of OC (Table 1). Meanwhile, organosulfates are estimated to contribute 1-2 % of PM$_{2.5}$ and 5-10 % OC in Eastern US (Shakya and Peltier, 2015). Therefore, the organosulfates quantified against authentic standards account for a minority of the total organosulfates, while other organosulfates likely comprise the majority of this class of compounds in Centreville, AL (as discussed in Sect. 3.3)."

This text has been revised to read: "The total contribution of the organosulfates quantified using authentic standards was less than 0.3 % of OC (Table 1). Meanwhile,  the estimated upper bound contribution of organosulfates to organic matter (OM) is 5.0 – 9.3 % in the SE US (Tolocka and Turpin, 2012).  Assuming OM/OC of 1.8 (Tolocka and Turpin, 2012), the calculated contribution of the organosulfates quantified  in this study  comprise 0.7 % of OM. Measurements of 2-methyltetrol sulfates reported by Ratanvahara et al. (2016) for Centreville had a mean concentration of 207.1 ng m$^{-3}$ and were estimated to account for 3.7%  while 2-methylglyceric acid sulfate had a mean concentration of 10.2 ng m$^{-3}$ and accounted for 0.2% of OM, y considering the average OC concentration of 3.07 ug m$^{-3}$ and an OM/OC ratio of 1.8. Together, the organosulfates quantified against authentic standards in Centreville accounts for 4.7 % of OM. Additional species that contribute significantly to MS$^2$ organosulfate signals are qualitatively and semi-quantitatively examined in Sect. 3.3.

**Anonymous Referee # 1 Weakness 5**: "*For your qualitative discussion of other major organosulfates present

*at CTR, what about OSs that do not fragment to the m/z 97 ion in MS²? Prior work has shown that other important organosulfates, especially from monoterpenes (like m/z 294), may produce only the m/z 96 product ion (Surratt et al., 2008, JPCA) in MS2 spectra. I would at least acknowledge that you may be missing some important organosulfates since you focus your analyses only on those that produce the m/z 97 product ion in MS² analyses."*

Response to Referee # 1 Weakness 5: We thank the referee for pointing this out. We have analyzed the organosulfates that fragmented to $m/z$ 96, but initially did not include these results because of the low signal (2 % of the precursors of $m/z$ 97). However, this comment suggests that the community would be interested in our findings from our studies of precursors to $m/z$ 96 and thus we have added them to the revised manuscript.

Results from $MS^2$ scans of precursors are shown in Figure S4 and Table S2.  In addition, these results have been discussed in added to section 3.3.  In particular, the following text has been added:

"In addition, a nitro-oxy organosulfate, $C_5H_{10}NSO_9^-$ (260.0076; Fig. S4) contributed up to 5.4 % of the $m/z$ 96 precursor ion signal (Table S1) and is also associated with isoprene (Surratt et al., 2008; Gómez-González et al., 2008)."

"Monoterpene-derived nitro-oxy organosulfates were particularly responsive to precursors of $m/z$ 96; $C_{10}H_{16}NSO_{10}^-$ (342.0495), $C_{10}H_{16}NSO_8^-$ (310.0597 (Ma et al., 2014; Surratt et al., 2008)) and $C_{10}H_{16}NSO_7^-$ (294.0647) (Table S1 and Fig. S4). The nitro-oxy organosulfate $C_{10}H_{16}NSO_7^-$ (294.0647) accounted for 25 % of the total $m/z$ 96 signal in $PM_{2.5}$ sample collected during nighttime on 10 July 2013 (Table S1). This semi-quantitative result is consistent with prior field studies reported $m/z$ 294 as the most abundant nitro-oxy organosulfate in SE US, particularly during night time (Gao et al., 2006; Surratt et al., 2008).

Other major organosulfate signals identified from $m/z$ 96, were $C_4H_7SO_4^-$ (151.0065), $C_3H_5SO_4^-$ (136.9909) and $C_5H_8NO_8S^-$ (241.9971), were not previously reported in the atmosphere (Table S2). Based on the molecular formula and double bond equivalence (Table S1), $m/z$ 151 is suggested as a methylallyl sulfate, $m/z$ 137 may be allyl sulfate and $m/z$ 242 may be a nitro-oxy organosulfate with a carbonyl group. However, the precursors to these organosulfates are unknown."

**Anonymous Referee # 1 Specific Comments:**

**Referee # 1 Specific Comment 1)** Abstract, Page 1, Lines 18-19: "*You should probably emphasize that this organosulfate is derived from multiphase chemistry of IEPOX (Surratt et al., 2010, PNAS; Lin et al., 2012, ES&T).*"

Response to Referee # 1 Specific Comment 1: While we agree with the reviewer, we do not think the abstract is the appropriate place to convey results from prior studies.  Instead, this information has been integrated into the introduction and discussion of 2-methyltetrol sulfate results.

**Referee # 1 Specific Comment 2)** Introduction, Page 2, Lines 2-5: "*Should you be more specific and emphasize that $PM_{2.5}$ has these adverse effects on human health and climate as well as contains most of the SOA?*"

Response to Referee # 1 Specific Comment 2: We agree with the reviewer and have revised the text accordingly.

The Introduction, Page 2, Lines 2-5 originally read: "Atmospheric particulate matter (PM) adversely affects human health and climate (Anderson et al., 2011; Kim et al., 2015; Rosenfeld et al., 2014; Levy et al., 2013). A significant fraction of PM is comprised of secondary organic aerosols (SOA) (Zhang et al., 2011) that form from reactions of volatile organic compounds (VOC) yielding semi-volatile products that partition to the aerosol phase."

This text has been revised to read: "Atmospheric fine particulate matter ($PM_{2.5}$; particles ≤2.5 μm in aerodynamic diameter) adversely affects human health (Valavanidis et al., 2008; Anderson et al., 2011; Kim et al., 2015) and influences the Earth's climate via direct and indirect radiative forcing (Novakov and Penner, 1993; Haywood and Boucher, 2000). A significant fraction of $PM_{2.5}$ organic matter is secondary in origin (Zhang et al., 2011), and forms by atmospheric oxidation reactions of volatile organic compounds (VOC) and partitioning of reaction products to the aerosol phase (Hallquist et al., 2009)."

**Referee # 1 Specific Comment 3)** Introduction, Page 2, Line 4: "*I would insert "atmospheric oxidation" before "reactions*""

Response to Referee # 1 Specific Comment 3) Introduction, Page 2, Lines 4: We have revised this sentence as indicated in the response to specific comment 2 (last sentence).

**Referee # 1 Specific Comment 4)** Introduction, Page 2, Lines 5-6: "*You should rephrase this sentence to be more correct. Maybe something like: "Organosulfates, which are produced from acid-catalyzed particle-phase reactions of gaseous oxidation products, such as epoxides (Lin et al., 2012, ES&T) and hydroperoxides (Mutzel et al., 2015, ES&T), contribute to SOA.*"

Introduction, Page 2, Lines 5-6 originally read: "Among SOA products are organosulfates, which are produced in the presence of sulfate aerosol and are particularly enhanced under acidic conditions (Surratt et al., 2007b; Surratt et al., 2010; Surratt et al., 2008; Surratt et al., 2007a)."

The text has been revised to read: "Among secondary organic aerosols (SOA) are organosulfates, which are mainly produced from acid-catalyzed particle-phase reactions of gaseous oxidation products such as epoxides (Lin et al., 2012) and hydroperoxides (Mutzel et al., 2015) with sulfate (Surratt et al., 2007b; Surratt et al., 2010; Surratt et al., 2008; Surratt et al., 2007a; Liao et al., 2015)."

**Referee # 1 Specific Comment 5)** Introduction, Page 2, Line 13: "*Now you switch to $PM_{2.5}$. You should define this since this is its first use.*"

Response to referee # 1 Specific Comment 5: We have implemented this suggestion and the revised text is provided in response to Referee # 1 Specific Comment 2.

**Referee # 1 Specific Comment 6)** Introduction, Page 2, Line 25: "*The beginning of this sentence should be reworded, possibly to "The most abundant organosulfates to be previously quantified include....."*"

Response to referee # 1 Specific Comment 6: We have revised this sentence as suggested.

**Referee # 1 Specific Comment 7)** Introduction, Page 2, Line 32: "*change "instead" to "used"*".

Response to referee # 1 Specific Comment 7: We have revised this sentence as suggested.

**Referee # 1 Specific Comment 8)** Introduction, Page 2, Line 32: "*Define the acronym "(-) ESI" for the first time here.*"

Response to Referee # 1 Specific Comment 8: Introduction, Page 2, Line 32: We have defined '(-) ESI', as indicated in the response to specific comment 7.

**Referee # 1 Specific Comment 9)** Introduction, Page 3, Line 9: "*change ", however" to "; however, "*"

Response to Referee # 1 Specific Comment 9: As suggested, we have changed the comma to a semicolon.

**Referee # 1 Specific Comment 10)** Introduction, Page 3, Lines 18-19: "*Not sure how relevant this sentence is to the discussion here. I believe the Ehn et al. (2010, ACP) study could measure extremely low vapor pressure products in the gas phase (there still of course is an equilibrium between the gas and aerosol phase) such as the glycolic acid sulfate due to the high sensitivity of their CIMS instrument.*"

Response to Referee # 1 Specific Comment 10: We agree with the referee that detection of gaseous glycolic acid sulfate in Ehn et al., 2010 emphasize the high sensitivity of their detection method (APi-ToF) to extremely low concentrations of glycolic acid sulfate in the gas phase that is in equilibrium with aerosol phase. Consequently, we have removed this sentence from the text.

**Referee # 1 Specific Comment 11)** Introduction, Page 3, Line 27: "*Change "epoxides" to "epoxydiols"*"

Response to Referee # 1 Specific Comment 11: We have revised this sentence as suggested.

**Referee # 1 Specific Comment 12)** Section 2.2: "*In this section, I would be clear on which samples were analyzed. You should also be clear on why on these samples were extracted and analyzed for this study.*"

Response to referee # 1 specific comment 12: As suggested by the reviewer, we have indicated which samples were analyzed for quantification of organosulfates and for the sampling artifacts study in section 2.2. The reason for why these samples were analyzed to identify major organosulfates in Centreville is given in response to weakness 2.

**Referee # 1 Specific Comment 13)** Page 7, Line 12: "*Is this an average glycolic acid sulfate concentration from this BHM study or the upper limit? Please clarify.*"

[revised manuscript text omitted]

SOAS 2013 Centreville Site Data Download, Meteorology Other, S-M01-ARA-MET: http://esrl.noaa.gov/csd/groups/csd7/measurements/2013senex/Ground/DataDownload/ access: 16th February, 2016, 2013.

Staudt, S., Kundu, S., Lehmler, H.-J., He, X., Cui, T., Lin, Y.-H., Kristensen, K., Glasius, M., Zhang, X., Weber, R. J., Surratt, J. D., and Stone, E. A.: Aromatic organosulfates in atmospheric aerosols: Synthesis, characterization, and abundance, Atmos. Environ., 94, 366-373, doi:http://dx.doi.org/10.1016/j.atmosenv.2014.05.049, 2014.

Stone, E. A., Hedman, C. J., Sheesley, R. J., Shafer, M. M., and Schauer, J. J.: Investigating the chemical nature of humic-like substances (HULIS) in North American atmospheric aerosols by liquid chromatography tandem mass spectrometry, Atmos. Environ., 43, 4205-4213, doi:http://dx.doi.org/10.1016/j.atmosenv.2009.05.030, 2009.

Surratt, J. D., Murphy, S. M., Kroll, J. H., Ng, N. L., Hildebrandt, L., Sorooshian, A., Szmigielski, R., Vermeylen, R., Maenhaut, W., Claeys, M., Flagan, R. C., and Seinfeld, J. H.: Chemical Composition of Secondary Organic Aerosol Formed from the Photooxidation of Isoprene, The Journal of Physical Chemistry A, 110, 9665-9690, doi:10.1021/jp061734m, 2006.

Surratt, J. D., Kroll, J. H., Kleindienst, T. E., Edney, E. O., Claeys, M., Sorooshian, A., Ng, N. L., Offenberg, J. H., Lewandowski, M., Jaoui, M., Flagan, R. C., and Seinfeld, J. H.: Evidence for Organosulfates in Secondary Organic Aerosol, Environ. Sci. Technol., 41, 517-527, doi:10.1021/es062081q, 2007a.

Surratt, J. D., Lewandowski, M., Offenberg, J. H., Jaoui, M., Kleindienst, T. E., Edney, E. O., and Seinfeld, J. H.: Effect of Acidity on Secondary Organic Aerosol Formation from Isoprene, Environ. Sci. Technol., 41, 5363-5369, doi:10.1021/es0704176, 2007b.

Surratt, J. D., Gomez-Gonzalez, Y., Chan, A. W. H., Vermeylen, R., Shahgholi, M., Kleindienst, T. E., Edney, E. O., Offenberg, J. H., Lewandowski, M., Jaoui, M., Maenhaut, W., Claeys, M., Flagan, R. C., and Seinfeld, J. H.: Organosulfate formation in biogenic secondary organic aerosol, J. Phys. Chem. A, 112, 8345-8378, doi:10.1021/jp802310p, 2008.

Surratt, J. D., Chan, A. W. H., Eddingsaas, N. C., Chan, M., Loza, C. L., Kwan, A. J., Hersey, S. P., Flagan, R. C., Wennberg, P. O., and Seinfeld, J. H.: Reactive intermediates revealed in secondary organic aerosol formation from isoprene, Proc. Natl. Acad. Sci., 107, 6640-6645, doi:10.1073/pnas.0911114107, 2010.

Tolocka, M. P., and Turpin, B.: Contribution of Organosulfur Compounds to Organic Aerosol Mass, Environ. Sci. Technol., 46, 7978-7983, doi:10.1021/es300651v, 2012.

Valavanidis, A., Fiotakis, K., and Vlachogianni, T.: Airborne Particulate Matter and Human Health: Toxicological Assessment and Importance of Size and Composition of Particles for Oxidative Damage and Carcinogenic Mechanisms, Journal of Environmental Science and Health, Part C, 26, 339-362, doi:10.1080/10590500802494538, 2008.

Xiong, F., McAvey, K. M., Pratt, K. A., Groff, C. J., Hostetler, M. A., Lipton, M. A., Starn, T. K., Seeley, J. V., Bertman, S. B., Teng, A. P., Crounse, J. D., Nguyen, T. B., Wennberg, P. O., Misztal, P. K., Goldstein, A. H., Guenther, A. B., Koss, A.

R., Olson, K. F., de Gouw, J. A., Baumann, K., Edgerton, E. S., Feiner, P. A., Zhang, L., Miller, D. O., Brune, W. H., and Shepson, P. B.: Observation of isoprene hydroxynitrates in the southeastern United States and implications for the fate of NOx, Atmos. Chem. Phys., 15, 11257-11272, doi:10.5194/acp-15-11257-2015, 2015.

Xu, L., Guo, H., Boyd, C. M., Klein, M., Bougiatioti, A., Cerully, K. M., Hite, J. R., Isaacman-VanWertz, G., Kreisberg, N. M., Knote, C., Olson, K., Koss, A., Goldstein, A. H., Hering, S. V., de Gouw, J., Baumann, K., Lee, S.-H., Nenes, A., Weber, R. J., and Ng, N. L.: Effects of anthropogenic emissions on aerosol formation from isoprene and monoterpenes in the southeastern United States, Proc. Natl. Acad. Sci., 112, 37-42, doi:10.1073/pnas.1417609112, 2015.

Zhang, Q., Jimenez, J. L., Canagaratna, M. R., Ulbrich, I. M., Ng, N. L., Worsnop, D. R., and Sun, Y.: Understanding atmospheric organic aerosols via factor analysis of aerosol mass spectrometry: a review, Anal. Bioanal. Chem., 401, 3045-3067, doi:10.1007/s00216-011-5355-y, 2011.

**Anonymous Referee # 2 Summary and Recommendations:** "*The manuscript by Hettiyadura et al. presents measured organosulphate (OS) concentrations in aerosol from the South East US from a four-day period during the SOAS campaign in the summer of 2013 at Centreville, Alabama. OS are an important contributor not necessarily due to their contribution to PM mass, but because they are the result of multi-phase processes and anthropogenic influence. The stated goals of the study are (i) quantification of OS (for which authentic standards are available) in $PM_{2.5}$, (ii) assessment of filter sampling artefacts, and (iii) identifying major OS in Centreville.*
*The analytical work is very thorough using state-of-the-art methods and the finding on the filter artefacts will be important for future work on OS. Similarly, the progress toward identifying/ruling out isomers/functional groups is an important contribution. The main concerns I have that should be addressed before publication is considered are clearer statements on the broader impact/significance beyond the analytical approach/work.*"

Response to the Referee # 2 Summary and Recommendations: We thank the referee for their review and suggestions. We have revised this paper carefully considering the referee's major comments, minor comments, technical comments and other comments. Our responses and revisions for each of the referee comment is provided point by point below.

**Anonymous referee # 2 Major Comment 1:** "*In order for the measurements to have significance beyond the very nice analytical method and artefact description and not simply be an anecdotal note of specific OS, it is critical to describe to what degree the very limited 4 day period was representative. As there was a plethora of measurements obtained at the Centreville site this should be easy. For example, where temperature, photochemical conditions, NOx conditions, amount of PM typical and, even more importantly, how variable were these conditions and is there any correlation with the observed OS variability shown in figure 1 (see point 2). Such a description of putting the measurements within the broader context of the SOAS campaign would help readers evaluate the broader significance of the observations described here.*"

Response to referee # 2 Major Comment 1: We agree with the referee that it is important to show how representative this subset of days to the larger SOAS.

The text at page 7, section 3.2, line 4 – 5 originally read: "The positive filter sampling artifacts associated with the three most abundant organosulfates quantified in Sect. 3.1 (glycolic acid sulfate, lactic acid sulfate and hydroxyacetone sulfate, respectively) were assessed."

This text has been revised to read: "The positive filter sampling artifacts associated with the three most abundant organosulfates quantified in Sect. 3.1 (glycolic acid sulfate, lactic acid sulfate and hydroxyacetone sulfate, respectively) were assessed from 07 – 11 July, 2013. This time period followed several days with rain, thus had slightly lower average $PM_{2.5}$ (5.24 ± 1.68 µg m$^{-3}$), OC (2.00 ± 0.67 µg m$^{-3}$), sulfate (1.26 ± 0.66 µg m$^{-3}$) and organosulfate concentrations relative to the average $PM_{2.5}$ (7.52 ± 3.41 µg m$^{-3}$), OC (3.07 ± 1.35 µg m$^{-3}$), sulfate (1.78 ± 0.81 µg m$^{-3}$) and organosulfate concentrations measured during SOAS in Centreville (Fig. 1 and Table 1). Within the studied subset of days, the 09 July daytime and nighttime,  and 10 July daytime concentrations (Fig. 1) were similar to the average conditions observed during SOAS, and are considered to be most representative of the average conditions at Centreville during SOAS."

This text has been added to section 3.3, page 9, line 8: "This analysis was applied to samples collected from 07 – 11 July, 2013, with a focus on the 10 July daytime sample with levels of $PM_{2.5}$ (7.01 ± 0.80 µg m$^{-3}$), OC (2.63 ± 0.21 µg m$^{-3}$), sulfate (1.06 ± 0.17 µg m$^{-3}$) and organosulfates (Fig. 1) near to the study average (Sect. 3.2 and Table 1)."

We have also extended the time series of organosulfate quantified to include 13 June – 13 July 2013 as shown in our new figure 1 and present correlation analysis in Table 2. The text that has been added to page 8, section 3.2:

"Correlations of hydroxyacetone sulfate, lactic acid sulfate and glycolic acid sulfate with co-located gas and aerosol measurements were used to gain insights to their potential precursors and conditions conducive to their formation (Table 2). Strong inter-correlations were observed for these organosulfates suggesting that they have common precursors and/or formation pathways. All three species had higher correlations with formaldehyde, MACR and glyoxal relative to isoprene, ISOPOOH and IEPOX that are low $NO_x$ oxidation products of isoprene (Bates et al., 2016; Krechmer et al., 2015)), as well as MVK and isoprene nitrates (ISOPN) that are high $NO_x$ oxidation product (Xiong et al., 2015)). While MVK, MACR, glyoxal and formaldehyde may be either biogenic or anthropogenic in origin, they primarily form from isoprene oxidation in SE US during summer (Xiong et al., 2015; Kaiser et al., 2015). Previous studies have shown that MVK, MACR, glyoxal and formaldehyde form in higher yields when isoprene was oxidized under high $NO_x$ (Kaiser et al., 2015; Liu et al., 2013). Of MVK and MACR, MACR is the major SOA precursor form from isoprene oxidation under high NOx conditions (Surratt et al., 2006; Kroll et al., 2006; Surratt et al., 2010). Thus the higher correlations with formaldehyde, MACR and glyoxal relative to other VOC precursors suggest that these organosulfates are enhanced by high $NO_x$ conditions.

All three species had moderate to strong correlations with sulfate, but not with liquid water content or acidity, suggesting that neither aerosol water nor aerosol acidity limit organosulfate formation. Similar correlations were reported at Centreville for isoprene derived SOA, and were attributed to variation of sulfate compared to consistently high aerosol acidity and high relative humidity observed during SOAS 2013 (Xu et al., 2015). Further, these correlations are consistent across other SOAS ground sites (Rattanavaraha et al., 2016; Budisulistiorini et al., 2015) indicating that the association of organosulfates with sulfate is a regional characteristic. The correlations of organosulfates derived from isoprene and sulfate in the SE US, suggests that sulfate is a key factor that influences biogenic SOA formation."

**Anonymous referee # 2 Major Comment 2:** "*It is stated that the work is complementary to that of Riva et al. 2016. However, it would be helpful if the similarities and differences with both the work/findings by Riva et al. 2016 and Rattanavaraha et al. 2016 would be stated more explicitly. For example, which OS were not observed by the two mentioned studies and how are the results similar and different? In fact, the time series in the Riva et al. 2016 (figure 4 of that manuscript) has a much longer dataset and*"

*it shows dramatic variability for organosulfate concentrations, which directly relates to point 1. In fact, a strong recommendation would be to collaborate with the Surratt group and use their much more extensive set of filter samples for the work described here.*"

Response to referee # 2 Major Comment 2: As suggested by the reviewer, we have discussed the relationship of our work to Riva et al. (2016) and Rattanavahara (2016) both quantitatively (as described in response to referee # 1 weakness 4) and qualitatively.

Page 7, section 3.1, lines 18 – 22 originally read: "The total contribution of the organosulfates quantified using authentic standards accounted for less than 0.5 % of $PM_{2.5}$ and less than 0.3 % of OC (Table 1). Meanwhile, organosulfates are estimated to contribute 1-2 % of $PM_{2.5}$ and 5-10 % OC in Eastern US (Shakya and Peltier, 2015). Therefore, the organosulfates quantified against authentic standards account for a minority of the total organosulfates, while other organosulfates likely comprise the majority of this class of compounds in Centreville, AL (as discussed in Sect. 3.3)."

This text has been revised to read: "The total contribution of the organosulfates quantified using authentic standards was less than 0.3 % of OC (Table 1). Meanwhile, the estimated upper bound contribution of organosulfates to organic matter (OM) is 5.0 – 9.3 % in the SE US (Tolocka and Turpin, 2012). Assuming OM/OC of 1.8 (Tolocka and Turpin, 2012), the calculated contribution of the organosulfates quantified in this study comprise 0.7 % of OM. Measurements of 2-methyltetrol sulfates reported by Ratanvahara et al. (2016) for Centreville had a mean concentration of 207.1 ng m$^{-3}$ and were estimated to account for 3.7% while 2-methylglyceric acid sulfate had a mean concentration of 10.2 ng m$^{-3}$ and accounted for 0.2% of OM, y considering the average OC concentration of 3.07 ug m$^{-3}$ and an OM/OC ratio of 1.8. Together, the organosulfates quantified against authentic standards in Centreville accounts for 4.7 % of OM. Additional species that contribute significantly to MS$^2$ organosulfate signals are qualitatively and semi-quantitatively examined in Sect. 3.3."

The text will be added at the end of section 3.3:
"The semi-quantitative results of organosulfates are both consistent and complementary to Riva et al. (2016) during SOAS. Five of the thirteen organosulfates quantified by Riva et al. (2016) in Centreville were among the ten major organosulfate signals observed herein; these included isoprene photo-oxidation products $C_5H_{11}SO_7^-$ (215.0225), $C_5H_9SO_7^-$ (213.0069), $C_3H_5SO_5^-$ (152.9858) and isoprene ozonolysis products $C_4H_7SO_6^-$ (182.9963) and $C_5H_{11}SO_6^-$ (199.0276). Other organosulfates, with *m/z* 181, 201, 227, 249, 267 and 315 were reported to have lower relative abundance (Riva et al., 2016) and were not among the ten major organosulfates in this study. Meanwhile, the organosulfate with *m/z* 197 ($C_5H_9SO_6^-$) was reported to be relatively high in Centreville (Riva et al., 2016), but was not identified as a major organosulfate in our study, likely due to differences in semi-quantitation methods. Together, these data demonstrate that organosulfates in Centreville are primarily derived from isoprene. In addition, our semi-quantiative analysis demonstrates relatively strong organosulfate signals from monoterpenes and to a lesser extent anthropogenic sources at Centreville."

**Anonymous referee # 2 Major Comment 3:** "*It is stated that the work provides new insights for the major OS species in the SE US. Again, it would be helpful to explicitly state what the new insights are. For example, which of the major OS had not been identified before, and if they had been identified it would be useful to describe what additional new insight is gained for each of the major species. Clearly, such new insights exist, e.g., resulting from the analytical approach such as ruling out carboxylic acid functional groups for some OS. Ending the manuscript by stating that there are new insights but not mentioning what they*

*are could then be improved. In summary, it would be helpful to make it easier for readers to identify clearly the novelty of the work/findings and the significance. To this end it may make sense to reorganize findings, e.g, (i) first show the 10 major OS, (ii) highlight the work to identify functional groups and isomers, which is a very nice and important contribution, and (iii) then discuss insights/recommendations. I think this may make it easier to recognize the significance, as the interesting new findings would not be interspersed within the "long" list of ten major OS. My second recommendation is to collaborate with the Surratt group using their extensive filter sample range, if possible."*

Response to anonymous referee # 2 Major Comment 3: We thank the referee for the helpful guidance provided to improve this section. We have reorganized the discussion of major organosulfates at section 3.3 as suggested, and summarized the new insights gain through semi-quantitative analysis at the end of the conclusion section as indicated below.

The text at section 3.3, page 10 line 5 - page 13 line 24 have been revised to read:

[revised manuscript text omitted]

**Minor comments:**

**Referee # 2 Minor Comment 1) p.1 line 5:** "*from biogenic volatile: : :*" As written it implies that only BVOCs form OS?"

Response to minor comment 1: We have deleted the word "biogenic" from this sentence to avoid misunderstanding.

**Referee # 2 Minor Comment 2) p.1 Line 13-14:** "*their VOC precursors*" is a little vague, as isoprene is one of the VOC precursors for OS, but I don't think the authors are implying that isoprene reacts on the filters to form SO. It would be useful to clarify."

Response to referee # 2 minor comment p.1 Line 13-4: We agree with the referee. We have replaced 'their VOC precursors' with 'gas phase precursors of organosulfates'.

**Referee # 2 Minor Comment 3) p.1 line 19:** "*Most of the ten: : :*" Please be specific. How many?"

The text at page 1, line 19 originally read: "Most of the ten most prevalent organosulfate were associated with biogenic VOC precursors (i.e. isoprene, monoterpenes, and 2-methyl-3-buten-2-ol [MBO])."

This text has been revised to read: "Nine of the ten strongest organosulfate signals were associated with biogenic VOC precursors (i.e. isoprene, monoterpenes, and 2-methyl-3-buten-2-ol [MBO])."

**Referee # 2 Minor Comment 4) p.2 line 2:** "*PM adversely affects : : : climate" This is a matter of debate. Some would say that PM positively affects climate due to counteracting greenhouse gas radiative effects. I would consider rephrasing.*"

Response to referee # 2 minor comment 4: We agree with the referee. We have rephrased this sentence in the response to referee # 1 specific comment 2 as given below.

Introduction, Page 2, Lines 2-5 originally read: "Atmospheric particulate matter (PM) adversely affects human health and climate (Anderson et al., 2011; Kim et al., 2015; Rosenfeld et al., 2014; Levy et al., 2013)."

This text has been revised to read: "Atmospheric fine particulate matter ($PM_{2.5}$; particles ≤2.5 μm in aerodynamic diameter) adversely affects human health (Valavanidis et al., 2008; Anderson et al., 2011; Kim et al., 2015) and influences the Earth's climate via direct and indirect radiative forcing (Novakov and Penner, 1993; Haywood and Boucher, 2000)."

**Referee # 2 Minor Comment 5) p.2 line 6-7:** "*The authors could also consider the work of Liao et al. 2015 as it discusses acid effects. Currently, only ground based studies are cited.*"

Response to referee # 2 minor comment 5: We have added this citation with the additional revisions made to this text in referee # 1 response to specific comment 4.

The text at page 2, line 5-7 originally read: "Among SOA products are organosulfates, which are produced in the presence of sulfate aerosol and are particularly enhanced under acidic conditions (Surratt et al., 2007b; Surratt et al., 2010; Surratt et al., 2008; Surratt et al., 2007a)."

The text has been revised to read: "Among secondary organic aerosols (SOA) are organosulfates, which are mainly produced from acid-catalyzed particle-phase reactions of gaseous oxidation products such as epoxides (Lin et al., 2012) and hydroperoxides (Mutzel et al., 2015) with sulfate (Surratt et al., 2007b; Surratt et al., 2010; Surratt et al., 2008; Surratt et al., 2007a; Liao et al., 2015)."

**Referee # 2 Minor Comment 6) p.2 line 12:** "*I think it would be more specific to state that OS may be useful markers for one type of anthropogenic influence on SOA formation from biogenic VOCs, as there surely must be aspects of anthropogenic influence that the sulphate does not represent.*"

Response to referee # 2 minor comment 6: We agree with the referee, we will indicate this in the text as given below.

The text at page 2, line 12 originally read: "Thus, organosulfates may be useful markers of anthropogenically influenced biogenic SOA."

This text has been revised to read: "Thus, organosulfates may be useful markers for sulfate-influenced biogenic SOA."

**Referee # 2 Minor Comment 7) p.3 line 28:** "*My understanding is that it is not clear whether the organosulfate is from methacrylic acid epoxide (MAE )or from hydroxymethyl-methyl-alpha-lactone (HMML), see Rattanavaraha et al. 2016?*"

Response to referee # 2 minor comment 7: We thank the referee for pointing this out; this text has been revised as follows:

This text has been revised to read: "2-Methylglyceric acid sulfate forms from either methacrylic acid epoxide (Lin et al., 2013) or hydroxymethyl-methyl-α-lactone (isoprene oxidation products), similarly to 2-methyltetrol sulfates, in the presence of sulfate under high NOx conditions (Nguyen et al., 2015)."

**Referee # 2 Minor Comment 8) p. 3 line 31-32:** "*I think follow-up studies (Gallowy et al. 2011 and Liao et al. 2015) showed that glycolic acid sulphate was unlikely to result from (photochemical formation) from glyoxal and that the mechanism/source was unknown? Similarly, the formation of lactic acid sulphate from methylglyoxal seems mechanistically challenging.*"

The text at page 3, line 31-32 originally read: "Formation of glycolic acid sulfate has been observed from reactive uptake of glyoxal to neutral or acidic sulfate aerosol upon irradiation (Galloway et al., 2009)."

This text has been revised to read: "Glycolic acid sulfate forms more efficiently from glycolic acid relative to glyoxal in the presence of acidic sulfate (Liao et al., 2015), while both precursors have biogenic and anthropogenic origins (Liao et al., 2015; Fu et al., 2008)."

The text at page 3, line 32 – 33 have been removed: "Lactic acid sulfate is also suggested to form from similar pathways from methylglyoxal (Shalamzari et al., 2013)."

**Referee # 2 Minor Comment 9) p. 4 line 25:** "*Please state the total organic carbon content as resistivity does not address the content of uncharged organic compounds.*"

The text at page 4, line 24-25 originally read: "Ultra-pure water was prepared on site (Thermo, Barnsted EasyPure-II; > 18.2 MΩ cm resistivity)."

This text has been revised to read: "Ultra-pure water was prepared on site (Thermo, Barnsted EasyPure-II; 18.2 MΩ cm resistivity, OC < 40 μg/L)."

**Referee # 2 Minor Comment 10) p. 5 line 7:** "*front QFF. Although it is fairly clear, defining better what the front QFF is would be useful (actual sample QFF?)*"

Response to referee # 2 minor comment 10: We have indicated that the front QFF is the filter that collect $PM_{2.5}$ during sampling in the original manuscript as given below.

The text at page 5, line 5-7 read: "Positive filter sampling artifacts associated with lactic acid sulfate, glycolic acid sulfate, and hydroxyacetone sulfate from 07-11 July 2013 were assessed using filter samples collected on bare back-up QFF ($Q_B$) and sulfuric acid impregnated back-up QFF ($Q_B$-$H_2SO_4$; $H_2SO_4$ - 8.65 μg cm$^{-2}$) collected in series behind front QFF ($Q_F$) that collected $PM_{2.5}$ (Fig. S1)."

**Referee # 2 Minor Comment 11) p. 6 line 29-30:** "*If the mass range was 400 Da, why consider up to 500 carbon atoms, corresponding to 6000 Da?*"

Response to referee # 2 minor comment p. 6 line 29-30: We agree with the referee that it is not necessary to use 500 carbons as the maximum mass range is 400 Da. These are the default settings used in the formula calculation software. In the future we will narrow down the range. However, it does not affect our results.

**Referee # 2 Minor Comment 12) p.7 line 18-20:** "*Please put these results in context with the ones previously mentioned by Tolocka and Turpin 2012).*"

Response to referee # 2 minor comment 12: The requested revision is provided in response to major comment 2.

**Referee # 2 Minor Comment 13) p.8 line 24-27:** "*I don't understand how the second sentence follows from the first: (i) there is some OS formation on the acidified filters, (ii) SOA is acidic enough and has high enough sulphate that these are not limiting factors. Are the authors implying that the gas-phase is already depleted of precursors or what is then limiting?*"

Response to referee # 2 minor comment 13: We thank the referee for pointing this out. Sulfate is the limiting factor in organosulfate formation, whereas other factors such as biogenic VOC precursors, aerosol acidity and aerosol water are consistently high in Centreville during SOAS, which is also consistent with the correlation results shown in response to major comment 1. The text at page 8, line 24 – 27 has been revised to read: "All three species had moderate to strong correlations with sulfate, but not with liquid water content or acidity, suggesting that neither aerosol water nor aerosol acidity limit organosulfate formation. Similar correlations were reported at Centreville for isoprene derived SOA, and were attributed to variation of sulfate compared to consistently high aerosol acidity and high relative humidity observed during SOAS 2013 (Xu et al., 2015). Further, these correlations are consistent across other SOAS ground sites (Rattanavaraha et al., 2016; Budisulistiorini et al., 2015) indicating that the association of organosulfates with sulfate is a regional characteristic. The correlations of organosulfates derived from isoprene and sulfate in the SE US, suggests that sulfate is a key factor that influences biogenic SOA formation."

**Referee # 2 Minor Comment 14) p. 9 line 3:** "*Does "negative sampling artefact" imply destruction of the OS in question? It would be helpful to clarify and explain.*"

Response to referee # 2 minor comment 14: We thank the referee for pointing this out and have clarified that negative sampling artifacts may "result from degradation during sampling, sample preparation, or analysis"

**Referee # 2 Minor Comment 15) p.9 line 23-24:** "*Does this mean that not all condensed-phase is evaporated when using water with little acetonitrile, i.e., that liquid water remains or just a few H₂O molecules on the OS and are such signals seen? Could it be that the water takes some of the charge and that or in some other way suppresses/reduces the ionization efficiency of the OS? Please explain this effect better.*"

Response to referee # 2 minor comment p.9 line 23-24: Experimentally we have seen decrease of MS signal response with increase of aqueouscomponent of the eluent, and *vice versa*. This is expected due to the low vapor pressure of water, relative to organic solvent such as acetonitrile, which suppress desolvation of ions in the ionization source. This will reduce the ions generate within the ESI source, thus result in lower signal. Organosulfates readily deprotonate in the ESI source (soft ionization technique). If assume that the water take some of the charge thus positively charged (abstracting $H^+$), yet they will be removed by the large negative potential applied in the ESI source, thus may not have an influence on deprotonated organosulfate ions.

The text at section 3.3, page 9, line 23 – 24 originally read: "Acetonitrile has a higher vapor pressure than water and more readily desolvates in the mass spectrometer, leading to higher signals."

This text has been revised to read: "Acetonitrile has a higher vapor pressure than water and more readily desolvates in the ionization source.  When increases the water content of the eluent, the signal of later-eluting ions is lower. Consequently, organosulfates retained longer on the BEH-amide column during HILIC gradient separation, such as organosulfates containing carboxyl and multiple hydroxyl groups are expected to be under-represented in this semi-quantitative analysis. These results emphasize the importance of using authentic standards to calibrate the instrument, particularly when using gradient elution."

**Referee # 2 Minor Comment 16) p.11 line 20:** "*Given the very high vapour pressure of MVK and MACR is it reasonable to assume that they are present in any significant concentration in PM in the first place to be able to oligomerize?*"

Response to referee # 2 minor comment 16: We agree that this is unlikely and have removed this sentence from the text.

**Anonymous Referee # 2 Technical Comments:** "*There are some grammatical errors, e.g., noun-verb agreement, and the manuscript could benefit from some proof reading.*"

**Response to Anonymous Reference # 2 Technical Comments:** We have addressed the technical comments point by point as indicated below.

**Technical Comment 1) p.2 line 13:** "*SOA accounts for a significant : : : and suggested: : :*""

The text at page 2, line 13 originally read: "SOA accounts for a significant fraction of organic $PM_{2.5}$ in SE US (Lee et al., 2010) and suggested to derive primarily from isoprene (Ying et al., 2015)."

This text has been revised to read: "SOA accounts for a significant fraction of organic $PM_{2.5}$ in SE US (Lee et al., 2010) and is expected to derive primarily from isoprene  (Ying et al., 2015)."

**Referee # 2 Technical Comment 2) p.2 Line 25:** "*Among them are most abundant organosulfate has been : : :*"

Response to referee # 2 technical comment 2: We have corrected this sentence in response to referee # 1 specific comment 6 as indicated below.

Introduction, Page 2, Line 24 - 31 originally read: "Among them are most abundant organosulfate has been 2-methyltetrol sulfate, followed by 2-methylglyceric acid sulfate, glycolic acid sulfate, lactic acid sulfate and hydroxyacetone sulfate during SOAS 2013 in Birmingham, AL (Rattanavaraha et al., 2016), Look Rock, TN (Budisulistiorini et al., 2015;  Riva et al., 2016) and Centreville, AL (Hettiyadura et al., 2015;  Riva et al., 2016)."

The text has been revised to read: "The most abundant organosulfates to be previously quantified, during SOAS 2013, using authentic standards include 2-methyltetrol sulfate (Budisulistiorini et al., 2015; Rattanavaraha et al., 2016), 2-methylglyceric acid sulfate (Budisulistiorini et al., 2015; Rattanavaraha et al., 2016), glycolic acid sulfate (Liao et al., 2015; Hettiyadura et al., 2015; Rattanavaraha et al., 2016), lactic acid sulfate (Hettiyadura et al., 2015) and hydroxyacetone sulfate (Budisulistiorini et al., 2015; Hettiyadura et al., 2015)."

**Referee # 2 Technical Comment 3) p.2 Line 31-32:** "*In the absence of authentic standards, surrogate standards are commonly instead, but can*"

Introduction, Page 2, Line 32 originally read: "In the absence of authentic standards, surrogate standards are commonly instead, but can lead to significant and often uncharacterized biases that result from differences in (-) ESI ionization efficiencies (Staudt et al., 2014)."

The text has been revised to read: "In the absence of authentic standards, surrogate standards are commonly used, but can lead to significant and often uncharacterized biases that result from differences in negative electrospray ionization ((-) ESI) efficiencies (Staudt et al., 2014)."

**Referee # 2 Technical Comment 4) p.3 line 9:** "*have been discussed", "are discussed" is perhaps more suitable*"

Introduction, Page 3, Line 9 originally read: "Thus MS$^2$ of precursors to bisulfate ion can be used for semi-quantification of organosulfates in the absence of authentic standards (Stone et al., 2009), however there are some limitations which have been discussed in Sect. 3.3."

The text has been revised to read: "Thus MS$^2$ of precursors to bisulfate ion (which scan all the precursors of a common product ion) can be used for semi-quantification of organosulfates in the absence of authentic standards (Stone et al., 2009); limitations of this approach are discussed in Sect. 3.3."

**Referee # 2 Technical Comment 5) p.8 line 20 14-15:** "*the potential : : : were assessed*"

The text at p.8 line 13-14 originally read: "The potential for glycolic acid sulfate, lactic acid sulfate and hydroxyacetone sulfate to form on QFF by acid catalyzed heterogeneous reactions were assessed..."

This text is revised to read: "The potential for glycolic acid sulfate, lactic acid sulfate and hydroxyacetone sulfate to form on QFF by acid catalyzed heterogeneous reactions was assessed...."

**Referee # 2 Technical Comment 6) p.11 line 19-21:** "*forms" instead of "form*"

Response to referee # 2 technical comment 6) p.11 line 19-21: This sentence has been removed in response to minor comment 16.

**Referee # 2 Technical Comment 7) p. 14 line 9**: "*of" instead of "to"?*"

The text at page 14, line 9 originally read: "The precursor ion scan to the bisulfate anion fragment (*m/z* 97) was used semi-quantitatively to assess major organosulfate species in ambient aerosol in the SE US."

This text will be revised to read: "The precursor ion scan of the bisulfate anion (*m/z* 97) and sulfate ion radical (*m/z* 96) were used semi-quantitatively to assess major organosulfate species in ambient aerosol in the Centreville, AL."

**Anonymous Referee Other Comments:** "*There are some places where it is not quite clear what is meant,*"

**Response to Anonymous Reference # 2 Other Comments:** We have addressed these comments point by point below.

**Referee # 2 Other Comment 1) p.2 line 16-17:** "*Stating that high sulphate etc. make the atmosphere subject to anthropogenic influence sounds a little odd to me. Do they actually not directly represent the anthropogenic influence?*"

Response to referee # 2 other comment 1: We thank the referee for pointing this out. We have revised this sentence as indicated below.

Introduction, Page 2, Line 13-21 originally read: "Together, high sulfate, isoprene, and aerosol acidity make the atmosphere in the SE US subject to anthropogenic influences on biogenic SOA formation (Weber et al., 2007; Goldstein et al., 2009; Watson et al., 2015)."

This text has been revised to read: "Together, high isoprene, sulfate and aerosol acidity make the SE US prime for the formation of sulfate-influenced biogenic SOA, including organosulfates."

**Referee # 2 Other Comment 2) p.8 line 9-11:** "*The very minor influence of : : : may be promoted : : :" I am not sure what promoting a minor influence means, and the "and possibly temperature" also seems a little out of place.*"

The text at section 3.2, page 8, line 9 - 12 have been removed: "The very minor influence of gas-phase glycolic acid sulfate and lactic acid sulfate in Centreville may be promoted by the higher organosulfate concentrations in the SE US, as well as the higher acidity (Guo et al., 2015) that can promote partitioning of acidic species like organosulfates to the gas-phase, and possibly temperature."

**Referee # 2 Other Comment 3) p. 3 line 8:** "*$MS^2$ has not been defined, I think. Some explanation of this method would be useful for readers to understand the following statements.*"

Response to Anonymous Referee # 2 other comment 3: We have provided a brief explanation to $MS^2$ of precursor ions in the response to technical comment 4.

iii. Multiple isomers of many organosulfates are observed with HILIC chromatography that co-elute under reversed-phase LC conditions. HILIC-MS/MS provides a basis for assessing the relative abundance of isomers and indicate that 1-hydroxybutane-3-one-2-sulfate is the dominant isomer of $C_4H_7SO_6^-$ (182.9963) in Centreville, AL. Likewise, $C_{10}H_{16}NSO_{10}^-$ (342.0495) and $C_7H_{11}SO_7^-$ (239.0225) are expected to be among abundant monoterpene derived organosulfates in Centreville, AL. Similar to Riva et al. (2016), $C_5H_{11}SO_6^-$ (199.0276) is relatively abundant in Centreville, but further experiments are need to identify its origin. Because of their relatively strong $MS^2$ signals, these species are also strong candidates for standard development and/or quantification in ambient aerosol.

Future efforts at standard development should focus on organosulfates that are expected to have high abundance, frequently detected in ambient aerosol, and/or have high specificity to VOC precursors.

**5 Data availability**

The SOAS research data used in this publication are available at http://esrl.noaa.gov/csd/groups/csd7/measurements/2013senex/Ground/DataDownload/.

**Acknowledgements**

The authors would like to thank E. Geddes, K. Richards, and T. Humphrey at Truman State University for synthesizing standards of benzyl sulfate, hydroxyacetone sulfate, and glycolic acid sulfate; S. Staudt at University of Wisconsin, Madison for synthesizing the lactic acid sulfate standard; L. Teesch and V. Parcell for their assistance in University of Iowa High Resolution Mass Spectrometry Facility (HRMSF); Ruikun Xin for his assistance in data processing; Eric Edgerton at Atmospheric Research & Analysis for providing $PM_{2.5}$ and sulfate measured in Centreville, AL during SOAS 2013; R. Weber, H. Guo and A. Nenes at Georgia Institute of Technology for providing aerosol acidity and aerosol water measured during SOAS 2013; P. Wennberg, T. Nguyen, J. St. Clair at California Institute of Technology for access to IEPOX, ISPOOH and ISOPN measurements; and A. Carlton from Rutgers University and J. Jimenez from University of Colorado, Boulder for organizing the SOAS component of Southeast Atmosphere Study. We would also like to thank US EPA Science To Achieve Results (STAR) program (grant number 83540101) for funding this research.

~~The authors would like to thank E. Geddes, K. Richards, and T. Humphrey at Truman State University for synthesizing standards of benzyl sulfate, hydroxyacetone sulfate, and glycolic acid sulfate; S. Staudt and F. Keutsch at University of Wisconsin, Madison (now at Harvard University) for synthesizing the lactic acid sulfate standard; L. Teesch and V. Parcell for their assistance in University of Iowa High Resolution Mass Spectrometry Facility (HRMSF); Ruikun Xin at University of Iowa (now at Carnegie Mellon University) for his assistance in data processing; Eric Edgerton at Atmospheric Research & Analysis for collecting providing $PM_{2.5}$ and meteorological data fromsulfate measured in Centreville, AL during SOAS 2013; R. Weber, H. Guo and A. Nenes at Earth and Atmospheric Science, Georgia Institute of Technology for providing aerosol acidity and aerosol water measured during SOAS 2013; P. Wennberg at California Institute of Technology, T. Nguyen (now at University of California, Davis) and J. St. Clair (now at University of Maryland, Baltimore County) for providing us the IEPOX, ISPOOH and ISOPN measurements in 
[revised manuscript text omitted]

---

## Author Response (AR2)

**Response to Editor Comments**

Journal: ACP

Title: Qualitative and Quantitative Analysis of Atmospheric Organosulfates in Centreville, Alabama

Author(s): Anusha P. S. Hettiyadura et al.

MS No.: acp-2016-636

MS Type: Research article

Iteration: Minor Revision

**Main Suggestion**: *Pg 8 line 25-pg 9 line 3: "I think this section would benefit from some more thought and analysis and I urge the authors to reconsider the reasoning in this section. In particular, I think that some of the correlations may be influenced by loss rates and lifetime issues. Please consider how formation, loss, and differing lifetimes may influence correlations and what this may tell us. Additionally, I would think that there would be a stronger correlation with isoprene nitrates if high NOx conditions enhanced formation. Finally how do the recent findings that MACR and MVK may be formed from reactions of isoprene hydroxy hydroperoxides on instrument surfaces (Rivera-Rios et al., 2014) influence this argument?"*

Response to main suggestion: We agree with the editor that the correlations can be influenced by the different life times of the organosulfates and VOC precursors and by artifacts and measurement uncertainties, particularly ISOPOOH, MACR, MVK and formaldehyde. We thank the editor for pointing this out. We have revised the paragraph on correlations of glycolic acid sulfate, lactic acid sulfate and hydroxyacetone sulfate with VOC precursors indicating the limitations associate with correlation analysis and emphasizing that further work is required to determine their VOC precursors and formation pathway.

The text at pg 8 line 25 – pg 9 line 3 originally read: "Correlations of hydroxyacetone sulfate, lactic acid sulfate and glycolic acid sulfate with co-located gas and aerosol measurements were used to gain insights to their potential precursors and conditions conducive to their formation (Table 2). Strong inter-correlations were observed for these organosulfates suggesting that they have common precursors and/or formation pathways. All three species had higher correlations with formaldehyde, MACR and glyoxal relative to isoprene, ISOPOOH and IEPOX that are low $NO_x$ oxidation products of isoprene (Bates et al., 2016; Krechmer et al., 2015)), as well as MVK and isoprene nitrates (ISOPN) that are high $NO_x$ oxidation product (Xiong et al., 2015)). While MVK, MACR, glyoxal and formaldehyde may be either biogenic or anthropogenic in origin, they primarily form from isoprene oxidation in SE US during summer (Xiong et al., 2015; Kaiser et al., 2015). Previous studies have shown that MVK, MACR, glyoxal and formaldehyde form in higher yields when isoprene was oxidized under high $NO_x$ (Kaiser et al., 2015; Liu et al., 2013). Of MVK and MACR, MACR is the major SOA precursor form from isoprene oxidation under high NOx conditions (Surratt et al., 2006; Kroll et al., 2006; Surratt et al., 2010). Thus the higher correlations with formaldehyde, MACR and glyoxal relative to other VOC precursors suggest that these organosulfates are enhanced by high $NO_x$ conditions."

This paragraph has been revised to read (pg 9 line 1 - 23 in the revised manuscript): "Correlations of hydroxyacetone sulfate, lactic acid sulfate and glycolic acid sulfate with co-located gas and aerosol measurements were used to gain

insights to their potential precursors and conditions conducive to their formation (Table 2). VOC measurements used for correlation analysis include ISOPOOH and IEPOX that are isoprene low NOx oxidation products (Paulot et al., 2009; Krechmer et al., 2015), formaldehyde, MACR, MVK and ISOPN that are isoprene high NOx oxidation products (Kaiser et al., 2015; Marais et al., 2016; Spaulding et al., 2003), hydroxyacetone and glycolaldehyde that are further oxidation products of MACR and MVK, respectively (Galloway et al., 2011; Spaulding et al., 2003), and glyoxal that form from both isoprene low and high NOx oxidation pathways (Marais et al., 2016). Organosulfates have longer lifetimes compared to isoprene and isoprene VOC precursors; glycolic acid sulfate and lactic acid sulfate reported to be stable for 21 days under highly acidic conditions (Olson et al., 2011), whereas above VOC have life times of several hours or less with respect to OH radicals which is their major sink (Paulot et al., 2009; Gaston et al., 2014; Lee et al., 2014; Montzka et al., 1993; Orlando et al., 1999; Lee et al., 1995). These differences in lifetimes confound correlation analysis, because their concentrations will vary on different time scales. Due to the longer lifetimes of organosulfates, they can also be transported to the sampling site from long distances affecting correlations with VOC precursors that are short lived thus locally formed. In addition, Rivera-Rios et al. (2014) has shown that MACR, MVK and formaldehyde form as artifacts from decomposition of ISOPOOH on metal surfaces during sampling and instrument analysis. Thus, there is a negative bias in the measurements of ISOPOOH and a positive bias in MACR, MVK and formaldehyde which may also influence their correlations. The strong inter-correlations observed for glycolic acid sulfate, lactic acid sulfate and hydroxyacetone sulfate suggest that they have common precursors and/or formation pathways. All three organosulfates correlated with both low and high NOx isoprene oxidation products. The relatively higher correlations of these organosulfates with formaldehyde, MACR and hydroxyacetone suggest that the high $NO_x$ pathway may play a larger role in their formation; however, this is not supported by lower correlations observed with ISOPN and MVK that are also high $NO_x$ isoprene oxidation products. Thus, further work is required to better understand the VOC precursors and formation pathways of these organosulfates and will likely require organosulfates with higher time resolution."

**Minor Suggestions:**

**Minor suggestion 1, pg 4 line 23**: "*should be sulfate radical ion*"

The text at pg 4 line 23 originally read: "The bisulfate anion ($HSO_4^-$ at *m/z* 97) and sulfate radical ($SO_4^{-\bullet}$ at *m/z* 96) is identified as a characteristic fragment ion of organosulfates (Gómez-González et al., 2008; Romero and Oehme, 2005)."

This text has been revised to read (pg 4 line 28 in the revised manuscript): "The bisulfate anion ($HSO_4^-$ at *m/z* 97) and sulfate radical anion ($SO_4^{-\bullet}$ at *m/z* 96) are characteristic fragment ions of organosulfates (Gómez-González et al., 2008; Romero and Oehme, 2005; Surratt et al., 2008)."

**Minor suggestion 2, pg 8 line 20-21**: "*please clarify if this is 3.7% of OM*"

Response to minor suggestion 2, pg 8 line 20 – 21: We agree that this is important to clarify.

This text has been revised to read (pg 8 line 26 – 30 in the revised manuscript): "Mean concentrations of 2-methyltetrol sulfates and 2-methylglyceric acid sulfate reported by Rattanavaraha et al. (2016) during SOAS 2013 in Centreville, AL were 207.1 ng m$^{-3}$ and 10.2 ng m$^{-3}$, respectively, which accounted for 3.7% and 0.2% of OM, for an average OC concentration of 3.07 µg m$^{-3}$ in Centreville during SOAS 2013 and OM/OC of 1.8."

**Minor suggestion 3, pg 8 line 21**: "*Please correct typo: "y" (end of the line)*"

Response to minor suggestion 3, pg 8 line 21: We have implemented this correction in the response to minor suggestion 2.

**Minor suggestion 4, pg 11 line 14**: ""*When increases the water content of the eluent…" Please revise*"

This text has been revised to read (pg 12 line 7 in the revised manuscript): "Thus, when increasing the water content of the eluent, the signal of later eluting ions decreases."

**Minor suggestion 5, pg 13 line 2**: "*Should this reference table S1?*"

Response to minor suggestion 5, pg 13 line 2 (pg 14 line 5 in the revised manuscript): Yes, it should be table S1, we have corrected this in the revised manuscript.

**Minor suggestion 6, pg 14 line 27**: ""*The precursors of bisulfate ion and sulfate radical insight…" I believe you are missing a word before "insight.""*

This text has been revised to read (pg 15 line 30 in the revised manuscript): "The precursors of bisulfate anion and sulfate radical anion provide insights…."

**References:**

Bates, K. H., Nguyen, T. B., Teng, A. P., Crounse, J. D., Kjaergaard, H. G., Stoltz, B. M., Seinfeld, J. H., and Wennberg, P. O.: Production and Fate of C4 Dihydroxycarbonyl Compounds from Isoprene Oxidation, The Journal of Physical Chemistry A, 120, 106-117, doi:10.1021/acs.jpca.5b10335, 2016.

[revised manuscript text omitted]